# PARP1 condensates differentially partition DNA repair proteins and enhance DNA ligation

Christopher Chin Sang [1,9], Gaelen Moore [1,9], Maria Tereshchenko [1,10], Hongshan Zhang [2,3,10], Michael L Nosella [1,4,10], Morgan Dasovich [5,7], T Reid Alderson [4,8], Anthony K L Leung [5,6], Ilya J Finkelstein [2,3], Julie D Forman-Kay [1,4] & Hyun O Lee [1✉]

## Abstract

Poly(ADP-ribose) polymerase 1 (PARP1) is one of the first responders to DNA damage and plays crucial roles in recruiting DNA repair proteins through its activity – poly(ADP-ribosyl)ation (PARylation). The enrichment of DNA repair proteins at sites of DNA damage has been described as the formation of a biomolecular condensate. However, it remains unclear how exactly PARP1 and PARylation contribute to the formation and organization of DNA repair condensates. Using recombinant human single-strand repair proteins in vitro, we find that PARP1 readily forms viscous biomolecular condensates in a DNA-dependent manner and that this depends on its three zinc finger (ZnF) domains. PARylation enhances PARP1 condensation in a PAR chain length-dependent manner and increases the internal dynamics of PARP1 condensates. DNA and single-strand break repair proteins XRCC1, LigIII, Polβ, and FUS partition in PARP1 condensates, although in different patterns. While Polβ and FUS are both homogeneously mixed within PARP1 condensates, FUS enrichment is greatly enhanced upon PARylation whereas Polβ partitioning is not. XRCC1 and LigIII display an inhomogeneous organization within PARP1 condensates; their enrichment in these multiphase condensates is enhanced by PARylation. Functionally, PARP1 condensates concentrate short DNA fragments, which correlates with PARP1 clusters compacting long DNA and bridging DNA ends. Furthermore, the presence of PARP1 condensates significantly promotes DNA ligation upon PARylation. These findings provide insight into how PARP1 condensation and PARylation regulate the assembly and biochemical activities of DNA repair factors, which may inform on how PARPs function in DNA repair foci and other PAR-driven condensates in cells.

**Keywords** PARP1; ADP-Ribosylation; Phase Separation; Biomolecular Condensates; DNA Repair
**Subject Category** DNA Replication, Recombination & Repair

## Introduction

Cells are frequently exposed to DNA-damaging agents such as reactive oxygen species or ionizing radiation that are detrimental to genome integrity. One of the first responders to DNA damage is poly(ADP-ribose) polymerase 1 (PARP1), the most abundant member of the human PARP family of proteins (Barkauskaite et al, 2015; Bai, 2015) that covalently modify biomolecules with one or more ADP-ribose (Leung et al, 2011; Altmeyer et al, 2015; Léger et al, 2014). PARP1 is allosterically activated (Eustermann et al, 2015) upon binding to single- and double-strand breaks in DNA (Eustermann et al, 2011; Langelier et al, 2011) and generates poly(ADP-ribose) or PAR on itself (Larsen et al, 2018; Jungmichel et al, 2013) and proteins in the vicinity of the DNA lesion, such as histones (Larsen et al, 2018; Karch et al, 2017). This localized PARylation leads to the recruitment of other proteins to the site of damage (Wei and Yu, 2016; Liu et al, 2017), including chromatin remodelers (Chou et al, 2010; Polo et al, 2010; Gottschalk et al, 2009) and DNA repair proteins (Aleksandrov et al, 2018; Koczor et al, 2021). Accordingly, PARP1 and its activity are crucial in multiple single- and double-strand break repair pathways (Ray Chaudhuri and Nussenzweig, 2017). Yet, how PARP1 influences the organization of damaged DNA and its target proteins and contributes to subsequent repair reactions is not well understood.

The enrichment of DNA repair proteins at DNA lesions, called DNA repair foci, has recently been described as biomolecular condensates (Patel et al, 2015; Altmeyer et al, 2015; Kilic et al, 2019; Levone et al, 2021; Li et al, 2022). Biomolecular condensates are non-membrane-bound compartments or biomaterials that can concentrate certain biomolecules and exclude others. Associative interactions between the molecules compensate for the entropic

[1]Department of Biochemistry, University of Toronto, Toronto, ON M5S 1A8, Canada. [2]Department of Molecular Biosciences, University of Texas at Austin, Austin, TX, USA. [3]Center for Systems and Synthetic Biology, University of Texas at Austin, Austin, TX, USA. [4]Molecular Medicine Program, The Hospital for Sick Children, Toronto, ON M5G 0A4, Canada. [5]Department of Biochemistry and Molecular Biology, Bloomberg School of Public Health, Johns Hopkins University, Baltimore, MD 21205, USA. [6]Department of Molecular Biology and Genetics, Department of Oncology, and Department of Genetic Medicine, School of Medicine, Johns Hopkins University, Baltimore, MD 21205, USA. [7]Present address: Green Centre for Reproductive Biology Sciences, University of Texas Southwestern Medical Centre, Dallas, TX, USA. [8]Present address: Institute of Structural Biology, Helmholtz Zentrum München, Munich, Bavaria, Germany. [9]These authors contributed equally: Christopher Chin Sang, Gaelen Moore. [10]These authors contributed equally: Maria Tereshchenko, Hongshan Zhang, Michael L Nosella. ✉E-mail: hyunokate.lee@utoronto.ca

cost of demixing, leading to their separation from the surrounding milieu as distinct phases in a process called phase separation (Hyman et al, 2014; Banani et al, 2017; Spannl et al, 2019). They are thought to regulate multiple biological processes, including ribosome biogenesis (Lafontaine et al, 2021; Correll et al, 2019), stress responses (Lau et al, 2020; Hofmann et al, 2021), and signal transduction (Su et al, 2021; Case et al, 2019) by concentrating specific components and influencing biochemical reaction rates (Banani et al, 2017; Lyon et al, 2021). Post-translational modifications drastically alter the assembly, composition, and material properties of condensates by modulating interactions between biomolecules, as shown for phosphorylation (Kim et al, 2019b; Tsang et al, 2019; Monahan et al, 2017), arginine methylation (Kim et al, 2019b; Nott et al, 2015; Qamar et al, 2018; Hofweber et al, 2018), and O-linked-N-acetylglucosaminylation (Nosella et al, 2021; Kim et al, 2021). The rapid and extensive recruitment and autoPARylation of PARP1 in response to DNA damage raises the possibility that it serves as a seed to nucleate DNA repair condensate formation.

Among the proteins recruited to DNA lesions by PARylation are enzymes involved in single-strand break repair (SSBR), a well-characterized pathway in which PARP1 acts as the primary sensor of single-strand breaks, including single-strand nicks. PARP1 activation at lesions leads to the recruitment of SSBR proteins; a scaffold protein XRCC1 interacts with DNA, PARP1, and PAR (Breslin et al, 2015; Polo et al, 2019; Kim et al, 2015; Mok et al, 2019; Pleschke et al, 2000) to bring together the proteins that will repair the break such as DNA polymerase β (Polβ) and DNA ligase III (LigIII) (Caldecott et al, 1994; Nash et al, 1997; Caldecott et al, 1996; Kubota et al, 1996). Polβ and LigIII also interact with PARP1 and PAR (Abbotts and Wilson, 2017; Caldecott, 2008), which likely contributes to their recruitment. Condensate-forming proteins such as FUS-EWS-TAF15 (FET) family proteins also localize to DNA damage sites in a PARP1 activity-dependent manner (Patel et al, 2015; Altmeyer et al, 2015; Rulten et al, 2014; Izhar et al, 2015) and play important roles in SSBR (Wang et al, 2018a). Whether and how PARP1 and its target proteins form DNA repair condensates and how their organization around damaged DNA is influenced by PARylation are unknown.

Here, we report that PARP1 forms condensates in a DNA-dependent manner. PARP1 autoPARylation enhances its condensation and differentially promotes the partitioning of single-strand break repair proteins, FUS, Polβ, LigIII, and XRCC1, within PARP1 condensates. Interestingly, PARP1 condensates concentrate short DNA, consistent with PARP1 clusters compacting long DNA and bridging DNA ends. Upon PARylation, PARP1 condensates further enrich the XRCC1-LigIII complex and enhance DNA ligation efficiency. Our findings support a model in which PARP1 nucleates condensates that selectively enrich and organize SSBR proteins at sites of DNA damage, promoting efficient ligation of DNA single-strand breaks following PARylation.

# Results

## PARP1 forms condensates in DNA-dependent manner

Human PARP1 (UniProt ID: P09874) comprises three regions—a zinc finger (ZnF) region, automodification domain (AD), and catalytic region (CAT; Fig. 1A)—containing several folded domains (Langelier et al, 2011; Tao et al, 2008; Hein et al, 2015) (PDB-IDs: 2COK, 2CR9) interspersed with regions of disorder that are 20-50 amino-acids long (Fig. EV1A). Three algorithms that predict the likelihood of phase separation based on either the sequence properties of intrinsically disordered protein regions or on sequence similarity to known RNA granule components differed dramatically in their estimates of whether PARP1 would undergo phase separation (Appendix Table S1). To test this experimentally, we examined the presence or absence of phase separation in solutions of recombinant mCherry-tagged human PARP1 in vitro (Fig. EV1B) using fluorescence microscopy. mCherry-PARP1 concentrations were varied over a range containing the estimated cellular concentration of 1–2 μM measured in HeLa cells (Hein et al, 2015). mCherry-PARP1 did not form condensates on its own at any of the tested protein and salt concentrations (Fig. 1B). However, the addition of a damaged DNA substrate—consisting of three oligonucleotides annealed to form a 50-nt triplex structure with a central nick and two blunt ends (triplex DNA; Appendix Table S2)—triggered the formation of micron-sized mCherry-PARP1 condensates. The presence of these condensates coincided with increasing PARP1 concentrations and lower salt concentrations (Fig. 1B,C quantified in D).

To further characterize the effects of DNA on PARP1 condensation, we quantified mCherry-PARP1 condensates in the presence of DNA of differing lengths and damage types by fluorescence microscopy. We found that the nicked triplex DNA promoted mCherry-PARP1 condensation in a concentration-dependent manner (Fig. 1E,F). Increasing the concentration ratio of PARP1 to DNA led to a reduction in PARP1 condensation (Fig. 1C, quantified in 1D). Similarly, at higher DNA to PARP1 ratios, PARP1 condensates did not form as readily (Fig. 1E, quantified in 1F), suggesting reentrant phase behavior (Milin and Deniz, 2018). PARP1 condensation was enhanced by increasing concentrations of a 25-nt oligonucleotide containing a single nick (nicked dumbbell DNA; Appendix Table S2) (Fig. EV1C) and double-stranded oligonucleotides with two blunt ends (dsDNA) (Figs. 1G and EV1D). Longer dsDNA promoted PARP1 condensation at lower DNA concentrations (Fig. 1H, left) and at higher salt concentrations compared to a shorter double-stranded DNA, or triplex DNA (Figs. 1H, right and EV1D compared to Fig. 1E). Interestingly, circular dsDNA with no broken ends or nicks (pUC19 plasmid, 2686 bp long) could also induce PARP1 condensate formation and did so to a similar extent with or without a single nick introduced by Nt.BspQI endonuclease (Figs. 1I and EV1E). Single-stranded DNA could also promote PARP1 condensate formation (Fig. EV1H,I), although at much higher concentrations than double-stranded DNA (Fig. 1G). Increasing the salt concentration blocked PARP1 condensate formation (Fig. EV1F). Up to 10% 1,6-hexanediol did not prevent PARP1 condensate formation, although it decreased condensate size (Fig. EV1G). These results are consistent with hydrophobic contributions being less significant in PARP1 condensate formation than other interactions, including electrostatics. Taken together, our data indicate that PARP1 undergoes condensation in a DNA-dependent manner and that DNA length strongly influences this process. Based on the estimated DNA footprint of PARP1 on DNA being 14 nucleotides flanking a single-strand break (Ménissier-de Murcia et al, 1989), we posit that a DNA fragment longer than 20

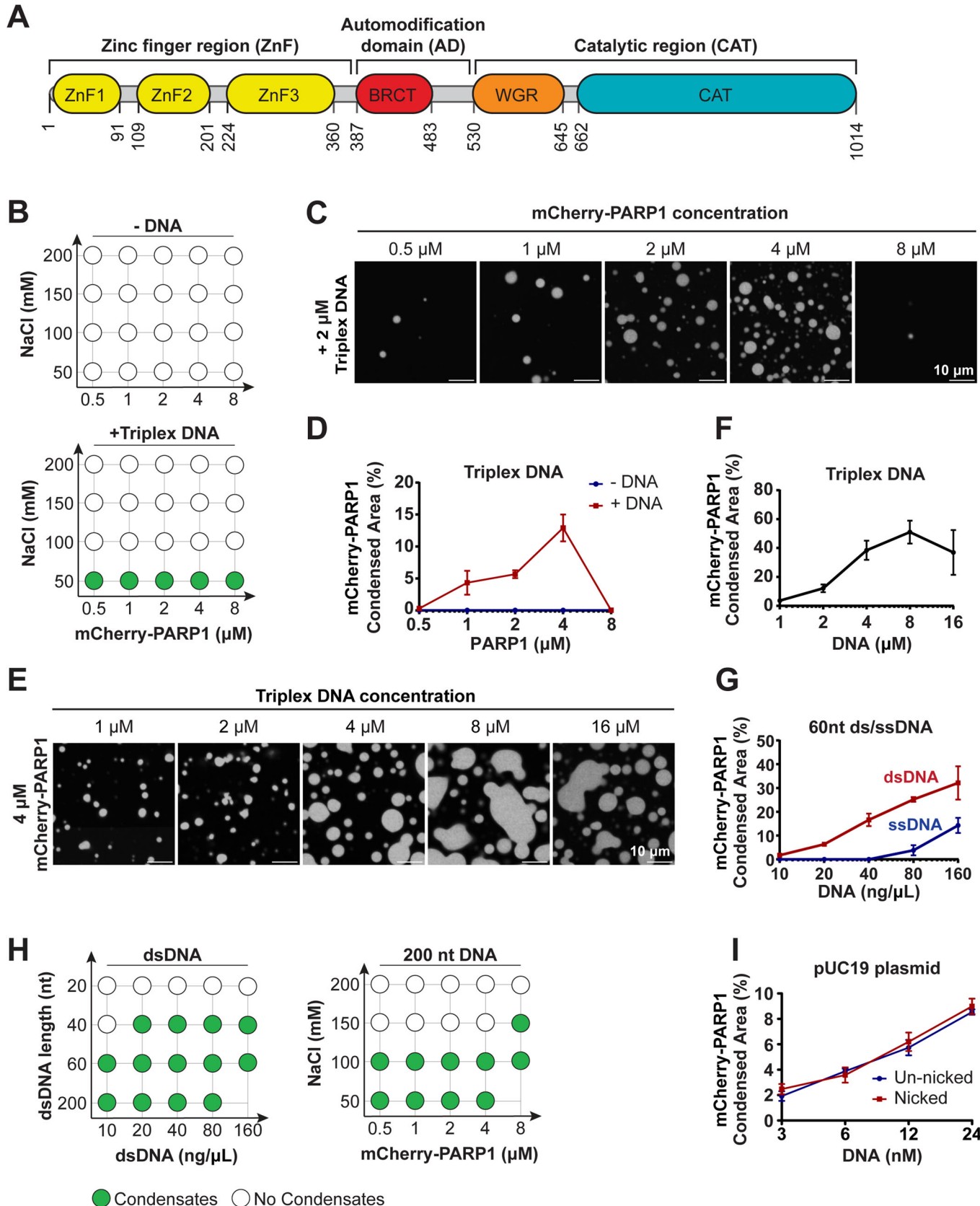

◀ **Figure 1. PARP1 forms condensates in a DNA-dependent manner.**

(A) Domain architecture of PARP1. ZnF zinc finger domain, BRCT BRCA1 C-terminal domain, WGR tryptophan-glycine-arginine region, CAT catalytic domain. (B) Phase diagram of condensates formed by recombinant mCherry-PARP1 at indicated concentrations with or without 2 μM triplex DNA. (C) Representative fluorescence micrographs of mCherry-PARP1 condensates diagramed in (B) at 50 mM NaCl. (D) Quantifications of the percent surface area covered by mCherry-PARP1 condensates in (B, C). (E) Fluorescence micrographs of 4 μM mCherry-PARP1 condensates with the indicated concentration of triplex DNA. (F) Quantifications of the percent surface area covered by mCherry-PARP1 condensates in (E). (G) Quantifications of the percent surface area covered by mCherry-PARP1 condensates with 60 nucleotide long (nt) single- or double-stranded DNA (dsDNA) at indicated DNA concentrations. (H) Phase diagram of condensates formed by (Left) 4 μM mCherry-PARP1 with the indicated dsDNA length and concentrations or by (Right) PARP1 with 20 ng/μL 200 nt DNA at indicated protein and salt concentrations. (I) Quantifications of the percent of surface area covered by mCherry-PARP1 condensates in the presence of pUC19 plasmid DNA with and without a single nick. All figures represent findings from at least three biological replicates in 20 mM Tris pH 7.5, 50 mM NaCl, 7.5 mM MgCl₂, and 1 mM DTT, unless otherwise indicated, with error bars indicating the standard error of the mean. Source data are available online for this figure.

nucleotides establishes the multivalency needed for PARP1 condensation.

## AutoPARylation enhances the formation and internal dynamics of PARP1 condensates

Upon binding to damaged DNA, PARP1 is allosterically activated, and PARylates both itself and other nearby proteins. To examine whether PARP1 autoPARylation influences its condensation, we activated the enzyme by mixing it with the nicked triplex DNA and the substrate for PARylation, the ADP-ribose donor nicotinamide adenine dinucleotide (NAD⁺) (Eustermann et al, 2015; Langelier et al, 2011, 2012, 2018; Dawicki-McKenna et al, 2015). This resulted in near-complete autoPARylation of PARP1 within 15 min in a DNA- and NAD⁺-dependent manner, as observed by its reduced electrophoretic mobility (Fig. 2A) and positive reactivity on an anti-PAR immunoblot (Fig. EV2A), consistent with previous reports (Eustermann et al, 2015; Langelier et al, 2011, 2012). PAR chain lengths were analyzed following cleavage and precipitation via chemoenzymatic labeling of terminal ADP-ribose moieties with Cy3-dATP and visualization on a urea-PAGE gel (Ando et al, 2019). PAR chains in our reaction ranged from 2 to 100+ ADP-ribose moieties, with the most abundant chain lengths ranging from ~10–60-mer (Fig. EV2C) at a total concentration of 35 to 5.9 μM, respectively (Fig. EV2B). This corresponds to an approximate concentration of 10 μM assuming a consistent length of 35 units. AutoPARylated mCherry-PARP1 formed larger condensates compared to unmodified mCherry-PARP1 condensates (Fig. 2B), while not significantly changing the conditions at which microscopically visible PARP1 condensation occurred in the presence of triplex DNA (Fig. EV2D compared to Fig. 1B). At lower PARP1 and DNA concentrations, where PARP1 does not form condensates, PARP1 condensate formation could be triggered in a PARylation-dependent manner (Fig. EV2E), suggesting that both DNA and PAR can modulate this process.

To determine how PAR chain length affects PARP1 phase separation, we limited the length of the chains on PARP1 by titrating the concentration of NAD⁺ or a competitive PARP1/2 inhibitor, ABT-888 (Penning et al, 2009), into the autoPARylation reaction mixture. Reducing the NAD⁺ concentration or increasing the ABT-888 concentration reduced autoPARylation of mCherry-PARP1 as observed by gel electrophoresis (Figs. 2C and EV2F). Reducing the NAD⁺ concentration also corresponded with diminished condensate size when visualized by fluorescence microscopy (Fig. 2D, corresponding images in Fig. EV2G). Increasing amounts of ABT-888 also decreased PARP1 condensate

size (Fig. EV2H). Additionally, adding a PAR-degrading enzyme, poly(ADP-ribose) glycohydrolase (PARG), reduced PAR chain lengths on mCherry-PARP1 and its condensation in a concentration-dependent manner (Fig. EV2I) and reduced the extent of PARP1 condensation back to the buffer only control (Fig. 2E, area quantified in Fig. 2F). PARP1 activity is often regulated by Histone PARylation Factor 1 (HPF1) (Rudolph et al, 2021b), which did not significantly influence PARP1 condensate size in the conditions tested (Fig. EV2J). Overall, these results suggest that autoPARylation enhances PARP1 condensation, which is proportional to the length of the PAR chains, and these effects can be regulated by PAR-modulating enzymes like PARG.

Non-PARylated PARP1 condensates triggered by DNA were spherical, suggesting that their morphologies were influenced by surface tension as commonly observed in liquids. However, the fluorescence recovery after photobleaching (FRAP) of a region within non-PARylated mCherry-tagged PARP1 condensates was limited, plateauing at ~25% normalized intensity, implying a mobile fraction of ~25% (Fig. 2G,H, −NAD⁺). This indicates that there is limited internal rearrangement and exchange between the condensate and the surrounding environment, as often seen in highly viscous condensates. Consistent with this, adjacent condensates of non-PARylated mCherry-PARP1 did not readily fuse or relax into spherical structures (Fig. 2I,J, −NAD⁺). In contrast, the fluorescence of autoPARylated mCherry-PARP1 condensates recovered steadily, reaching close to complete recovery levels over time (Fig. 2G,H, +NAD⁺). In addition, adjacent autoPARylated mCherry-PARP1 condensates fused readily, relaxing into a spherical shape within 2 min of touching (Fig. 2I,J, +NAD⁺). A similar trend was observed using dumbbell DNA (Fig. EV2K), suggesting that PARP1 condensate dynamics are consistent across different types of DNA substrates. Together, these data indicate that autoPARylation increases the mobility of molecules within PARP1 condensates.

## The ZnF region is required for PARP1 condensation

Given the importance of DNA for PARP1 condensation, we asked if the zinc finger region (ZnF), which binds DNA, is sufficient and necessary for this process. We generated two ZnF truncation mutants, PARP1-ΔZnF, which includes all regions of PARP1 except for the three ZnF domains, and PARP1-ZnF, which only contains the three ZnFs and excludes all other regions (Figs. 3A and EV3A). We analyzed their phase separation alone or together at various protein concentrations in the absence or presence of triplex DNA and NAD⁺. In the absence of DNA, neither of the truncation

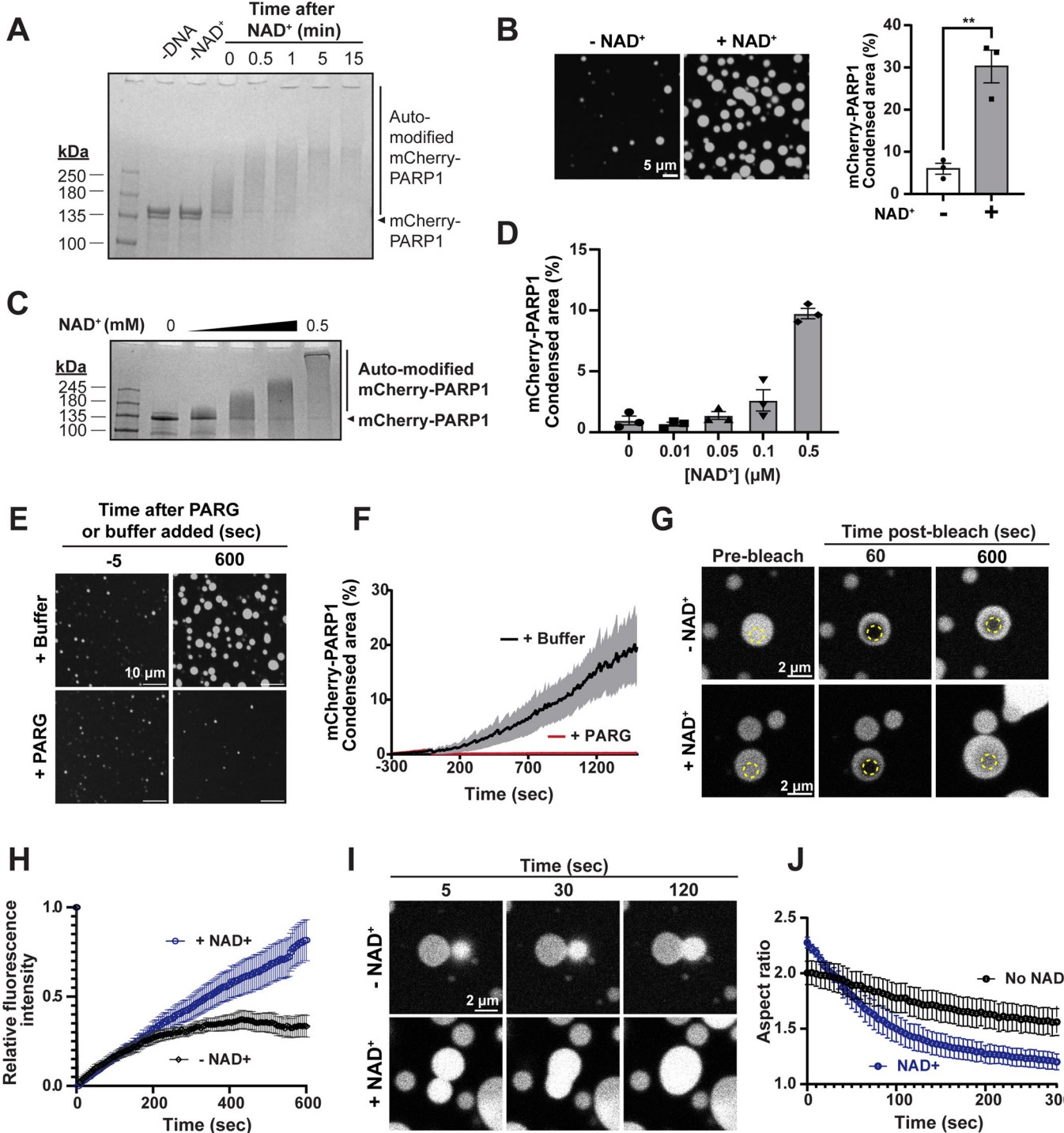

mutants (PARP1-ΔZnF and PARP1-ZnF) condensed at concentrations as high as 32 μM (Fig. 2B, corresponding images in Fig. EV3B,C). Upon the addition of DNA, PARP1-ZnF readily formed condensates at concentrations as low as 8 μM (Fig. 3B,C), whereas PARP1-ΔZnF was unable to form condensates at any tested concentration (Figs. 3B and EV3C). Deletion of either the ZnF region (PARP1-ΔZnF) or the remainder of the protein (PARP1-ZnF) abolished the autoPARylation activity of PARP1

(Fig. EV3D), consistent with previous reports (Langelier et al, 2011, 2012; Ali et al, 2012), and accordingly showed no NAD$^+$-dependent change in condensation of the truncations (Fig. 3C). These findings indicate that the interactions between the ZnF region and DNA are important for PARP1 condensation and may be sufficient to drive the process at higher concentrations.

To examine whether the linkage between the ZnF region and the rest of the PARP1 is important for condensate formation,

Figure 2. AutoPARylation enhances the formation and internal dynamics of PARP1 condensates.

(A) AutoPARylation assay: SDS-PAGE of 1 µM mCherry-PARP1 at indicated times before or after the addition of 0.5 mM NAD$^+$ or of 0.3 µM dumbbell DNA. AutoPARylation leads to smearing and reduced electrophoretic mobility of mCherry-PARP1. (B) Left: Fluorescence micrographs of 4 µM mCherry-PARP1 with or without autoPARylation with 0.5 mM NAD$^+$ and 1.2 µM triplex DNA. Right: Quantifications of the percent of the surface area covered by mCherry-PARP1 condensates. $P$ values are obtained from a student $T$-test: **$p < 0.01$ ($p$ value = 0.0040). (C) AutoPARylation assay of 1 µM mCherry-PARP1 with 0.3 µM dumbbell DNA and increasing concentrations of NAD$^+$ (0, 0.01, 0.05, 0.1, and 0.5 mM). (D) Quantifications of the percent surface area covered by 1 µM mCherry-PARP1 condensates with 0.3 µM triplex DNA at indicated NAD$^+$ concentrations. (E) Fluorescence micrographs of 1 µM mCherry-PARP1 condensates formed in the presence of 0.05 µM triplex DNA and 0.5 mM NAD$^+$. At 5 min of incubation, either 0.5 µM PARG or PARylation buffer were added. (F) Quantifications of the percent surface area covered by mCherry-PARP1 condensates in (E). (G) Fluorescence recovery after photobleaching (FRAP): micrographs of condensates formed by 4 µM mCherry-PARP1 and 1 µM triplex DNA with and without NAD$^+$. Condensates were photobleached in the region indicated by the dotted circles, and fluorescence recovery was monitored over time. (H) FRAP quantifications represented in (G). $n = 20$ from three biological replicates. (I) Fluorescence micrographs showing fusion of condensates formed by 4 µM mCherry-PARP1 and 1 µM triplex DNA condensates with and without 0.5 mM NAD$^+$. Time from initiation of fusion events is indicated. (J) Droplet aspect ratio plot over time for five independent fusion events. All figures represent findings in PARylation Buffer (20 mM Tris pH 7.5, 50 mM NaCl, 7.5 mM MgCl$_2$, 1 mM DTT) from at least three biological replicates unless specified otherwise. Error bars indicate the standard error of the mean. Source data are available online for this figure.

PARP1-ZnF was mixed with PARP1-ΔZnF. An equimolar mixture of the PARP1 truncations restored condensation at concentrations as low as 4 µM (Fig. 3D), which is lower than the concentration needed for ZnF-only condensates (8 µM) (Fig. 3C) but higher than the concentration at which full-length (FL)-PARP1 condenses (1 µM) (Fig. 3E) in the presence of DNA. Mixtures of PARP1 truncations alone did not form condensates in the absence of DNA at all concentrations tested, just like FL-PARP1 (Fig. 3F, corresponding images in Fig. EV3E,F). Interestingly, autoPARylation was restored when PARP1-ΔZnF and PARP1-ZnF were mixed at equimolar concentrations as low as 1 µM (Fig. EV3D) but did not significantly enhance condensation of the mixture in the conditions tested (Fig. 3D). This suggests that the interactions between the ZnF and ΔZnF in cis are important for PARP1 condensation in the presence of DNA, more so than their PARylation status.

Next, we investigated whether the linkage of three ZnF domains in the ZnF region is important for condensation. We treated PARP1 with either caspase-3 WT, which cleaves PARP1 between ZnF2 and ZnF3 during apoptosis (Lazebnik et al, 1994), or caspase-3 C163S, a catalytically-dead mutant (Fig. 3G). Treating mCherry-PARP1 with caspase-3 WT abolished condensation, irrespective of autoPARylation, while treatment with caspase-3 C163S did not affect mCherry-PARP1 condensation (Fig. 3H). These results highlight that the three ZnFs in tandem are essential for PARP1 condensation. Together, the interaction of the ZnF region with the rest of the protein drives PARP1 condensation with DNA.

## PARP1 compacts long DNA and bridges DNA ends

To understand the interaction of PARP1 and damaged DNA within condensates, we mixed mCherry-PARP1 with Cy5-labeled triplex DNA. In the absence of NAD$^+$ (and thus no PARylation), Cy5-triplex DNA was enriched ~20-fold in PARP1 condensates when compared to the concentration in the dilute phase (Fig. 4A, −NAD$^+$), as determined by the fluorescence intensity within condensates compared to that of the dilute phase. In the presence of NAD$^+$ (and thus with PARylation), Cy5-triplex DNA in PARP1 condensates was enriched ~12-fold compared to the dilute phase (Fig. 4A, +NAD$^+$), which was less than that of pre-PARylation.

Next, we examined how PARP1 influences longer DNA via single-molecule DNA curtains (Zhang et al, 2023). In this assay, a 48.5 kb DNA substrate (λ-phage DNA) is tethered to the passivated surface of a microfluidic flowcell via a biotin-streptavidin linkage.

The DNA is stained with the intercalating dye SYTOX Orange. After injecting PARP1, DNA molecules immediately compacted in a protein concentration-dependent manner (Figs. 4B–D and EV4A). The DNA compacted by 92.2 ± 3.1% (Mean ± SD) to 94.4 ± 1.8% when 1–20 nM of WT PARP1 was injected in the flowcell ($N > 48$ for all conditions). The rate of DNA compaction was dependent on the concentration of PARP1, ranging from 0.18 ± 0.11 kb/s for 1 nM PARP1 to 2.73 ± 0.49 kb/s for 20 nM PARP1 ($N > 48$ DNA molecules for all conditions). In contrast, a mutant missing the ZnF domains (ΔZnF) does not compact DNA at 20 nM, and neither does the ZnF domain mutant alone at this concentration (Fig. EV4B). A PARylation-deficient mutant PARP1-E988Q (Marsischky et al, 1995; Rolli et al, 1997) compacted DNA at similar rates (Fig. 4C). The ionic strength of the buffer solution significantly influenced DNA compaction by PARP1. 20 nM PARP1 was sufficient to compact DNA in a buffer with 25 mM NaCl (Fig. 4C), but no compaction was observed in a buffer with 150 mM NaCl (Fig. EV4C,D). Increasing PARP1 concentration to 400 nM resulted in DNA compaction even in the presence of 150 mM NaCl (Fig. EV4C). Lastly, we quantified the effect of PARylation on PARP1-DNA condensates by chasing the PARP1 injections with varying concentrations of NAD$^+$. Washing the flowcell with 500 µM NAD$^+$ buffer resulted in 70% of the DNA extending to their full lengths (Fig. 4E,F) and higher and lower NAD$^+$ concentrations increased or decreased the speed at which this occurred, respectively, consistent with previous findings with magnetic tweezers (Bell et al, 2021). As expected, the 500 µM NAD$^+$ wash did not reverse DNA compaction in the presence of PARylation-deficient mutant PARP1-E988Q (Fig. EV4E,F). These results demonstrate that PARP1–DNA interactions leading to DNA compaction are sensitive to ionic strength, do not require PARylation, but can be modulated by the modification.

Interestingly, injection of a sub-saturating 100 µL of 10 nM PARP1 into the flowcell led to end-to-end bridging between two adjacent DNA molecules (Fig. 4G; Movie EV1). The bridged DNA ends withstood applied forces of ~0.17 ± 0.02 pN, as estimated by using the Worm-like chain model (Bell et al, 2021; Schaub et al, 2018). To monitor the effect of PARylation on ends-bridged DNA, we introduced 500 µM NAD$^+$ into the imaging buffer. Adding NAD$^+$ resulted in the resolution of DNA bridges with wild-type PARP1 (Fig. 4G; Movie EV1). PARylation-deficient mutant PARP1-E988Q could also bridge DNA ends, but adding NAD$^+$ did not influence its behavior as expected (Fig. EV4G; Movie EV2).

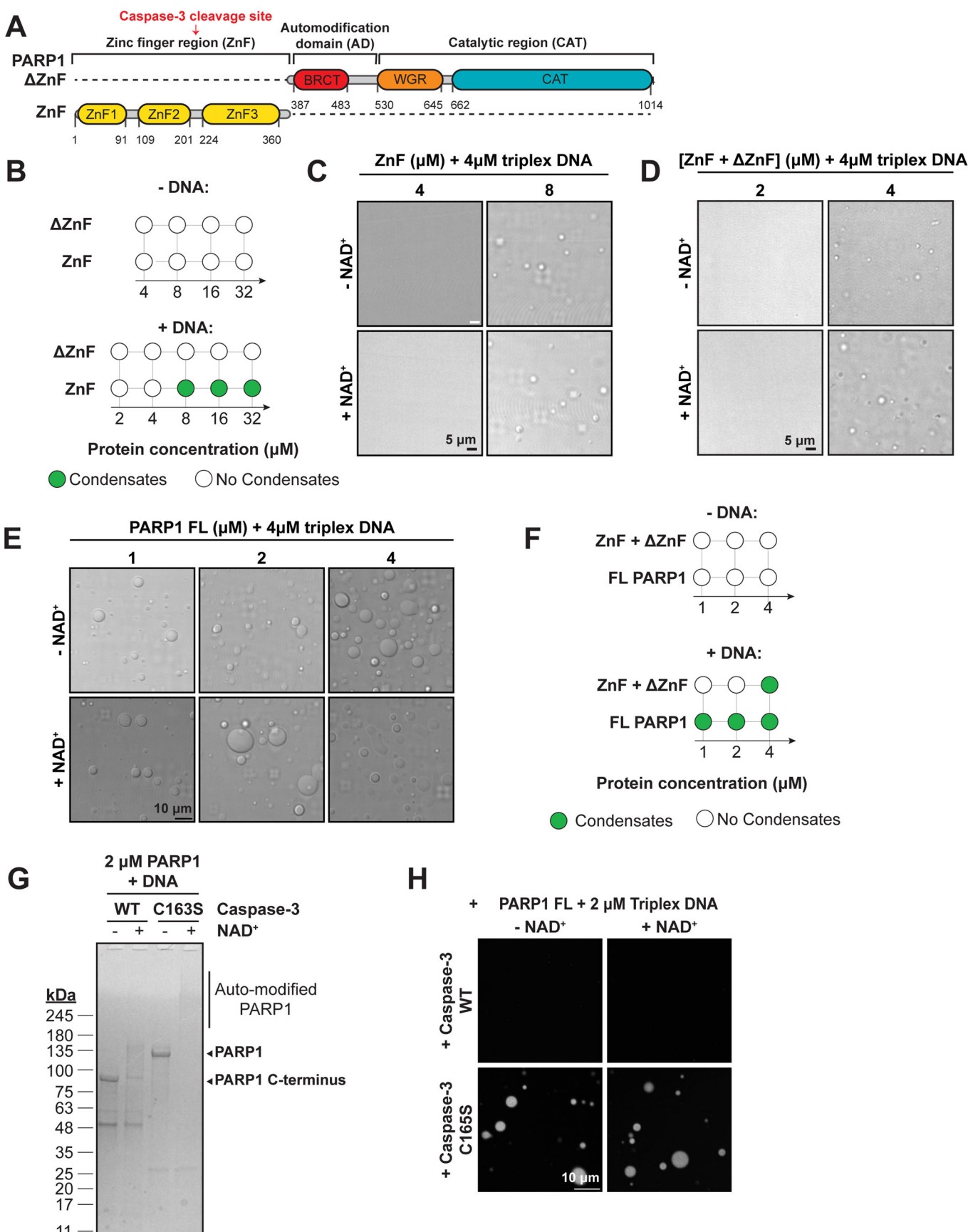

◀

**Figure 3. The ZnF region is important for PARP1 condensation.**

(A) Domain architecture of truncated PARP1 proteins. (B) Phase diagram of condensates formed by truncated PARP1 proteins at the indicated concentrations. Note that the phase diagrams did not change with or without NAD$^+$. (C) DIC micrographs of ZnF PARP1 protein with 4 μM triplex DNA. (D) DIC micrographs of truncated PARP1 proteins with 4 μM triplex DNA with or without 0.5 mM NAD$^+$. (E) DIC micrographs of full-length PARP1 proteins with 4 μM triplex DNA with or without 0.5 mM NAD$^+$. (F) Phase diagram of condensates formed by both truncated proteins mixed together or full-length PARP1 at the indicated concentrations. Note that the phase diagrams did not change with or without NAD$^+$. (G) Gel-based autoPARylation assay of mCherry-PARP1 with wild-type caspase-3 or catalytically inactive caspase-3 C163S. 1 μM wild-type or C163S caspase-3 was added to 2 μM mCherry-PARP1 and incubated for 30 min. After incubation, the indicated samples were incubated with 0.5 mM NAD$^+$ for 15 min before analysis. (H) Fluorescence micrographs of 2 μM mCherry-PARP1 with 2 μM triplex DNA, and with or without NAD$^+$, pre-cleaved with 1 μM wild-type or C163S caspase-3. Caspase-3 was added to mCherry-PARP1 30 min prior to NAD$^+$ addition to ensure full PARP1 cleavage, and images were obtained 10 min after cleavage reaction initiation. All figures represent findings from at least three biological replicates in 20 mM Tris pH 7.5, 50 mM NaCl, 7.5 mM MgCl$_2$, and 1 mM DTT unless specified otherwise. Source data are available online for this figure.

To further characterize this phenomenon, we designed an oligonucleotide capture assay (Fig. EV4H). Here, a mixture of 5 nM PARP1 and 20 nM ATTO647N-labeled 60 bp dsDNA was introduced into the flowcell and incubated for 5 min with buffer flow turned off. Subsequently, flow resumed, and images were acquired (Fig. EV4I). All PARP1-DNA clusters were co-localized with DNA-ATTO647N, indicating that PARP1 could bridge interactions between the oligonucleotide and long DNA substrate. Adding 500 μM NAD$^+$ resulted in 98% of the 60 bp oligos dissociating from the DNA condensate (Fig. EV4I,J). Together, we conclude that PARP1 can bridge DNA substrates in *trans*, and that these bridges can be reversed by autoPARylation.

## PARP1 condensation and activity enhances nicked DNA ligation

A possible function of PARP1 condensation is to organize and concentrate DNA repair enzymes and their substrates. To test this, we investigated how PARP1 phase separation contributes to the organization of several SSBR proteins: the DNA ligase LigIII, its obligatory binding partner XRCC1, the DNA polymerase Polβ, and FUS (Fig. EV5A). When mixed with mCherry-PARP1, AlexaFluor488-labeled XRCC1 and LigIII were heterogeneously distributed within PARP1 condensates individually (Figs. 5A and EV5B, −NAD$^+$) and together (Fig. 5D, −NAD$^+$). The heterogeneous partitioning of XRCC1 and LigIII did not change upon PARylation (Figs. 5A,D and EV5B, +NAD$^+$). PARylation did enhance the overall enrichment of LigIII and XRCC1 in PARP1 condensates approximately twofold (Fig. 5A–C). Interestingly, when LigIII and XRCC1 were added together, PARP1 demixed from the XRCC1/LigIII complex and DNA, such that areas enriched in mCherry-PARP1 were depleted in triplex DNA and LigIII whereas LigIII-rich areas strongly partitioned damaged DNA while depleting mCherry-PARP1 (Fig. 5D). This behavior is reminiscent of multiphase condensates in which two or more phases coexist within a condensate, such as in nucleoli (Lafontaine et al, 2021; Feric et al, 2016). XRCC1 and LigIII were readily PARylated (Fig. EV5C) in the presence of PARP1 and NAD$^+$.

Another SSBR protein, AlexaFluor488-labeled Polβ, was evenly enriched in the PARP1 condensates (~1.5-fold) (Fig. EV5D, −NAD$^+$), and its partitioning in PARP1 condensates did not change with PARylation (Fig. EV5D, +NAD$^+$). FUS-GFP was also evenly enriched within PARP1 condensates (~1.5-fold) (Fig. EV5E, −NAD$^+$), but its partitioning was significantly increased (~4-fold) upon PARylation (Fig. EV5E,F, +NAD$^+$). Based on previous reports that XRCC1 and FUS strongly interact with PAR (48, 71),

their increased partitioning in PARylated PARP1 condensates may be driven by PAR binding. Interestingly, strong enrichment of FUS following PARylation increased the internal dynamics of PARP1 condensates. In the absence of PARylation, when FUS is weakly enriched, FUS did not impact PARP1 dynamics (Fig. EV5G). However, with PARylation, FUS accelerates the recovery time of PARP1 within condensates (Fig. EV5G), indicating that the presence of FUS enhances the dynamic properties of PARP1 condensates. Together, our data indicate that PARP1 condensates and PARylation differentially organize SSBR proteins XRCC1, LigIII, Polβ, and FUS, and certain molecules like FUS can influence the dynamics of PARP1 condensates.

Next, we investigated how PARP1 condensation influences DNA repair efficiency by monitoring the ligation of a nicked DNA substrate in the presence of XRCC1, LigIII, and PARP1 with and without NAD$^+$ over time. Interestingly, in the presence of PARP1 condensates (triggered with NAD$^+$ addition and thus PARylation), ligation increases nearly threefold (Fig. 5E). At low PARP1 concentrations in which microscopically visible condensates do not form before or after NAD$^+$ addition, there were no significant differences in ligation rates with or without PARP1 or PARylation (Fig. EV5H,I). These results suggest that the organization of PARP1 into condensates, rather than the presence of PARP1 or PARylation alone, may promote efficient DNA ligation, potentially by enriching LigIII, XRCC1, and damaged DNA upon PARylation.

Lastly, to better understand the progression of the ligation reaction, we performed time-lapse imaging of PARP1 condensates with XRCC1, LigIII, and damaged DNA. PARP1 initially forms condensates that enrich triplex DNA, as in Fig. 4A, but when XRCC1 and LigIII are added, and form multiphase condensates, triplex DNA becomes progressively more enriched in the XRCC1 and LigIII phase (Fig. 5F; Movie EV3). This phenomenon occurred similarly with or without PARylation (Fig. EV6A, +NAD$^+$), and was driven strongly by LigIII and less so by XRCC1 (Fig. EV6B). The multiphase organization did not significantly change with the addition of PARG (Fig. EV6C), consistent with the phenomenon being independent of PARylation. Together, these results suggest that the preferential binding of damaged DNA to the LigIII enzyme is sufficient to drive the hand-off of damaged DNA between PARP1 and LigIII in multiphase condensates.

## Discussion

In this study, we examine the role of PARP1 and PARylation in the formation, organization, and function of nascent, multi-component

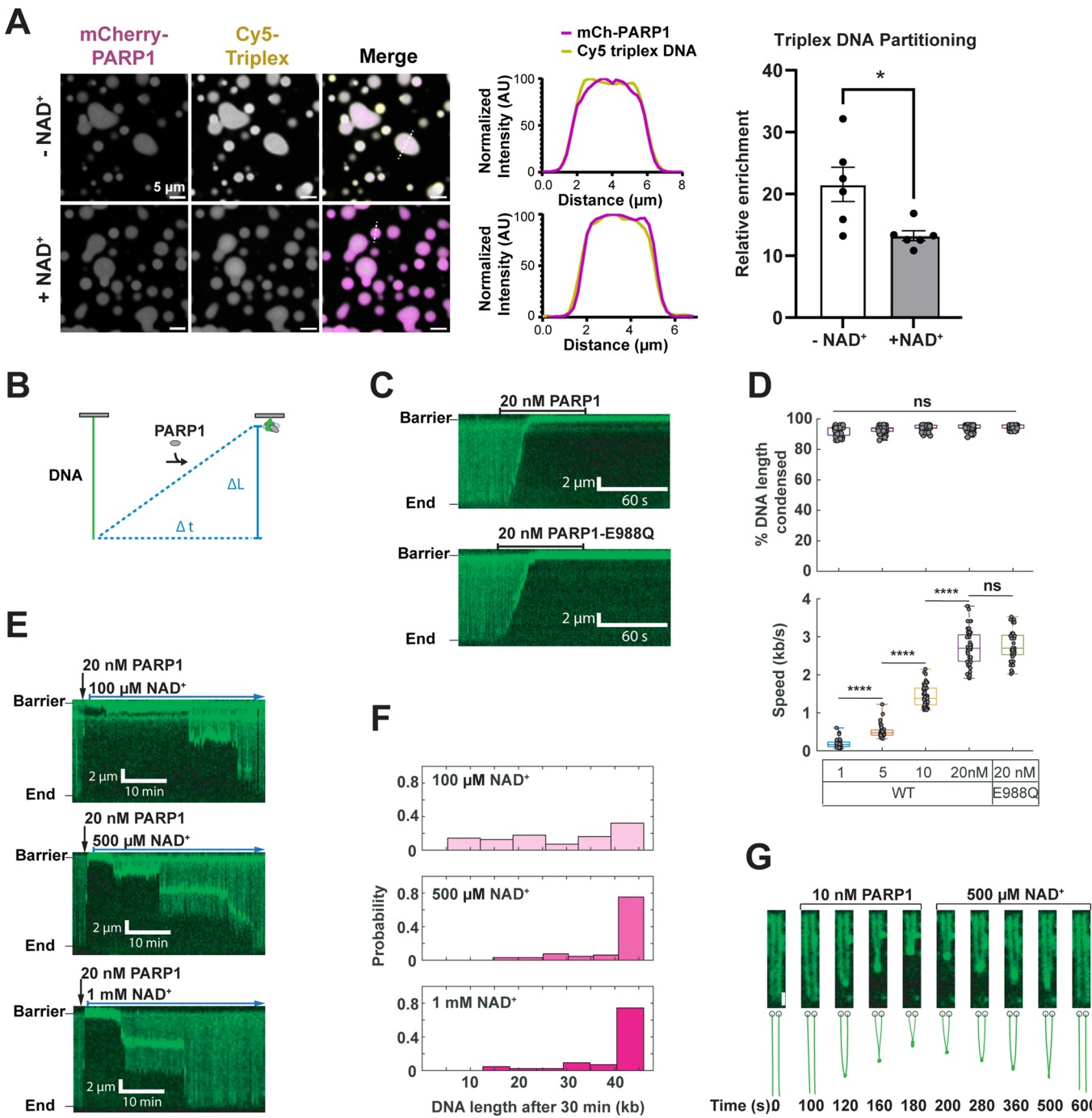

DNA repair condensates in vitro. We report that PARP1 readily forms viscous condensates in a manner dependent on the concentration of DNA and PAR. PARP1 autoPARylation enhances the formation and dynamics of PARP1 condensates and has differing effects on the organization of DNA repair proteins within them. Functionally, PARP1 condensates concentrate short DNA, which correlates with PARP1 clusters compacting and bridging long DNA ends in a single-molecule DNA curtains assay. Furthermore, the activity of PARP1 condensates enhances DNA ligation. Together, these findings suggest a model whereby

autoPARylation of PARP1 seeds the formation of condensates at DNA lesions that support efficient DNA repair (Fig. 6A).

PARP1 is an interesting example of a predominantly folded protein that undergoes condensation. Unlike many other phase-separating proteins such as the FUS-like RNA-binding proteins, which have disordered regions hundreds of residues long (Wang et al, 2018b), PARP1 has only short ~10–20 residue-disordered regions interspersed between multiple highly structured interaction domains. PARP1 does not form condensates on its own but requires the presence of DNA to do so (Fig. 1). This is similar to

**Figure 4. PARP1 condensates bridge broken DNA ends.**

(A) Fluorescence micrographs of 4 μM mCherry-PARP1 and 4 μM Cy5-triplex DNA with and without NAD$^+$ in 20 mM Tris pH 7.5, 50 mM NaCl, 7.5 mM MgCl$_2$, and 1 mM DTT. The fluorescence intensity measured across the diameter of a representative condensate (dashed white line) is plotted in the center. Quantification of Cy5-triplex enrichment in mCherry-PARP1 condensates is shown on the left. Enrichment is calculated as the fluorescence intensity in the condensed phase relative to the dilute phase; $N = 6$ biological replicates. Error bars indicate the standard error of the mean. $P$ value is obtained from a student's $t$-test; *$p < 0.05$ ($p$ value = 0.0290). (B) Schematic of DNA condensation initiated by PARP1. The change in DNA length (ΔL) over a time (Δt) is used to calculate the mean DNA compaction rate (ΔL/Δt). (C) Representative kymographs showing DNA (green) compaction after injecting 100 μL of 20 nM of the indicated PARP1 variant. Both WT and the PARylation-defective PARP1-E988Q can condense DNA. (D) DNA compaction rates are dependent on the PARP1 concentration, with complete compaction observed at all tested concentrations. Boxplots denote the first and third quartiles of the data. The center lines in the box indicate the median. The upper whisker is the maximum value and the lower whisker is the minimum value. At least 49 DNA molecules were analyzed for each condition. $p$ values are obtained from a two-tailed $t$-test: ****$p < 0.0001$, ns not significant. The $p$ values (from left to right) are 6.7015e-15, 2.3758e-18, and 1.0076e-18, respectively. (E) Injecting NAD$^+$ reverses DNA compaction. Black arrow: injection of 20 nM WT PARP1. Blue arrow: continuous injection of the indicated NAD$^+$ concentration. (F) Histogram of DNA lengths after they were first incubated with 20 nM WT PARP1 and then washed with NAD$^+$ for 30 min. $N > 43$ DNA molecules for each condition. (G) Frames from a movie showing the end-to-end bridging of two DNA molecules (green) following the injection of 100 μL of 10 nM PARP1. The bridging event can be reversed by injecting 500 μM NAD$^+$. Scale bar: 2 μm. Source data are available online for this figure.

other nucleic acid-binding proteins, such as G3BP1 (Yang et al, 2020) and VRN1 (Zhou et al, 2019), whose condensation requires the presence of RNA and DNA, respectively. Multivalent associative interactions drive condensate formation and nucleic acids add to the multivalency of the system through interactions with each other as well as bringing bound proteins into closer proximity. In our system, the multivalency of the system is further increased by the three tandem ZnF domains in PARP1 (Fig. 3) and by the PAR chains, which are nucleic acid-like polymers of up to 200 ADP-ribose moieties (Barkauskaite et al, 2015; Bai, 2015; Alvarez-Gonzalez and Jacobson, 1987). Our finding that ZnF domains are essential for PARP1 condensation is interesting as other regions of PARP1 have been shown to also interact with DNA (Rudolph et al, 2021a; Chappidi et al, 2024). The strength of interactions between DNA and PARP1 regions may contribute to these differences. Supporting this notion, the first two ZnF domains, which have been shown to directly sense and bind single-strand nicks with high affinity (Eustermann et al, 2015; Ali et al, 2012; Deeksha et al, 2023), were necessary for PARP1 condensation (Caspase-3, Fig. 3). However, further increasing DNA concentration dissolves PARP1 condensation, possibly because PARP1 undergoes a reentrant phase transition (Fig. 1). RNA-binding proteins, including FUS, hnRNPA1, TDP-43, have been shown to display similar reentrant phase behavior (Maharana et al, 2018), as the protein-protein interactions are diluted by increased protein-RNA interactions. Many different types of DNA, such as double-stranded DNA with and without nicks and blunted ends, as well as single-stranded DNA to a lesser extent, could trigger PARP1 condensation. Longer length and double-strandedness were strong promoters of this process; length likely contributes to the multi-valency of the system, as does the stronger affinity of double-stranded DNA to PARP1, when compared to single-stranded DNA (Deeksha et al, 2023). In addition to nucleosome-free DNA, our recent work indicates that PARP1 can form condensates with nucleosome-bound damaged DNA as well (Nosella et al, 2024). PARP1 interacts with single- and double-stranded DNA breaks (Deeksha et al, 2023). Consistent with this, we found that both double- and single-stranded DNA induce PARP1 condensates, as does intact circular DNA plasmid, although in decreasing order of propensity. Thus, PARP1 condensates may form in many contexts with DNA, although the extent of condensation or its impact on enzymatic activity will likely differ and should be evaluated for each context. For example, PARP1 may not condense as readily on

mismatches, bulky adducts, and other damaged DNA, as it does on double- and single-stranded breaks. In addition to DNA damage, increasing evidence points towards PARP1's role in transcriptional regulation (Gibson et al, 2016; Huang and Kraus, 2022). Elucidating how cells limit the propensity for PARP1 to form condensates to certain DNA regions would be an interesting area of further research.

Our finding that autoPARylation enhances the formation of PARP1 condensates (Fig. 2) provides further evidence to support the hypothesis that PARylated proteins act as seeds for condensate formation in cells (Altmeyer et al, 2015; Singatulina et al, 2019; Leung, 2020). Free PAR chains enhance in vitro condensate formation of various RNA-binding proteins, such as the FET proteins, hnRNPA1, and TDP-43 (Patel et al, 2015; Duan et al, 2019; McGurk et al, 2018; Rhine et al, 2022). Our study shows that PARP1 autoPARylation enhances its condensation, and PAR chain length is crucial in this process, with short PAR chains promoting condensation much less effectively than long chains. In addition, PARylation increased the internal mobility of PARP1 condensates in our study. This is in contrast to other studies that showed free PAR chains reduced the internal dynamics of other condensates (Duan et al, 2019) and suggest that PAR may have differing effects on different condensates. Given the increased dynamics of PARP1 condensates upon PARylation, it is tempting to speculate that PARylated PARP1 condensates may be easier to disassemble. In addition, higher concentrations of PAR may dissolve PARP1 condensates, similar to PARP1 reentrant behavior with high concentrations of DNA. Consistent with this idea, a recent study reported that PARylation disassembled PARP1 condensates (Chappidi et al, 2024). Thus, PARP1 condensation and its dynamics may be regulated by the lengths and/or concentration of PAR chains. Investigating other properties of PAR, such as branching (Alvarez-Gonzalez and Jacobson, 1987), should provide further insights into how PARylation regulates PARP1 condensates. Recent findings suggest that two other PARP family members, the catalytic domain of PARP5a and full-length PARP7, form condensates in vitro upon ADP-ribosylation (Rhine et al, 2022; Zhang et al, 2020), suggesting that many PARPs may form condensates, although possibly involving different domains, with or without (poly)ADP-ribosylation or interactions with nucleic acids.

We show that PARP1 condensates concentrate ~50 nt-long DNA. Similarly, clusters of PARP1 that form at much lower concentrations than the micron-sized PARP1 condensates can

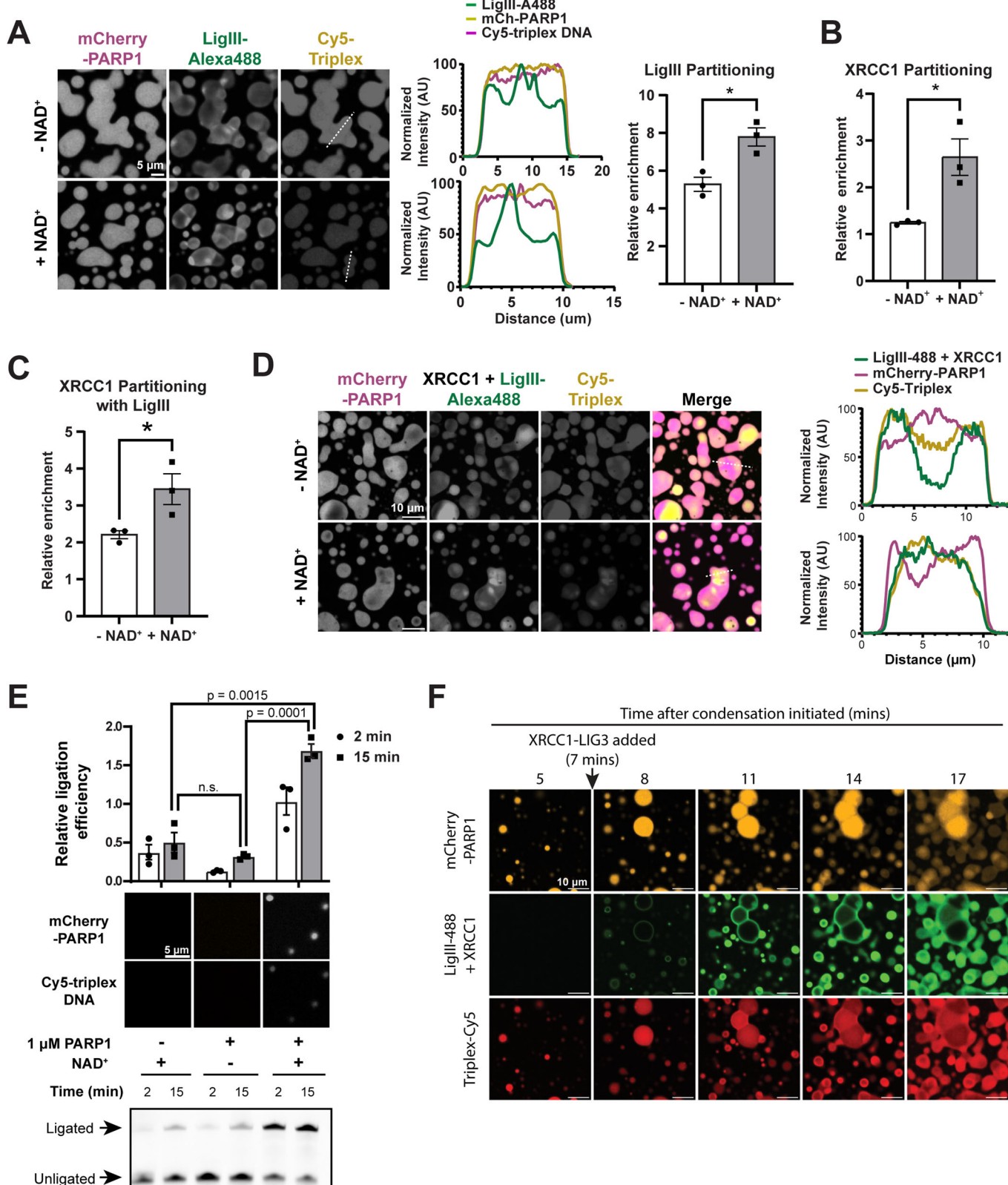

◀ **Figure 5. SSBR proteins partition together in condensates that enrich DNA.**

(A) Left: Fluorescence micrographs of 4 µM mCherry-PARP1, 4 µM triplex DNA, and 1 µM LigIII (10% AlexaFluor488-labeled) with and without NAD$^+$. The fluorescence intensity measured across the diameter of a representative condensate (white dashed line) is plotted in the middle. Right: Quantification of the relative enrichment of LigIII within condensates as calculated by the ratio between the average intensity within the condensed phase versus the dilute phase. *p* value = 0.0147. (B) Quantification of the partitioning of 1 µM XRCC1 within 4 µM mCherry-PARP1 with 4 µM triplex DNA with and without NAD$^+$, as in (A). *p* value = 0.0233. (C) Quantification of the partitioning of 1 µM XRCC1 within 4 µM mCherry-PARP1 with 1 µM LigIII, 4 µM triplex DNA with and without NAD$^+$. *p* value = 0.0477. (D) Fluorescence micrographs of 4 µM mCherry-PARP1, 4 µM triplex DNA, 1 µM LigIII (10% AlexaFluor488-labeled), and 1 µM XRCC1 with NAD$^+$. The fluorescence intensity measured across the white dashed line is plotted on the right. (E) Bottom: Gel-based ligation assay performed with 10 nM LigIII, 10 nM XRCC1, 50 nM Cy5-triplex DNA and with or without 1 µM mCherry-PARP1 and 0.5 mM NAD$^+$, analyzed 2 or 15 min after ATP addition. Top: Quantifications of ligation efficiency calculated as the fluorescence intensity of Cy5-triplex DNA in the ligated band versus the unligated bands (representative gel image on the bottom). Middle: Fluorescence micrographs of mCherry-PARP1 and Cy5-triplex DNA in the ligation reactions after ATP addition. (F) Fluorescence micrographs of 4 µM mCherry-PARP1 with 4 µM Triplex DNA, and 1 µM XRCC1 and LigIII complex. Imaging started 2 min after PARP1 and DNA were mixed, and XRCC1 and LigIII mixture was added 5 min after imaging started. All figures represent findings from at least three biological replicates in 20 mM Tris pH 7.5, 50 mM NaCl, 7.5 mM MgCl$_2$, and 1 mM DTT unless specified otherwise. Scale bar = 10 µm. Error bars indicate the standard error of the mean. *p* values are obtained from a student's *t*-test: *$p < 0.05$, n.s. non-significant. Source data are available online for this figure.

compact long DNA of over 48,000 base pairs. This is consistent with results obtained with atomic force microscopy that PARP1 can cluster DNA (Bell et al, 2021). Our data that PARP1 clusters can bridge long DNA ends, which is consistent with recent findings (Chappidi et al, 2024), also raises the possibility that PARP1 condensates may bridge, and potentially protect, naked and broken DNA ends. Protecting and bridging broken DNA ends to keep them in close proximity would be a crucial first step to efficient DNA repair in multiple repair pathways (Cannan and Pederson, 2016). A recent study reported that a condensate-forming protein, FUS, can compact and bridge DNA ends (Renger et al, 2022), similar to PARP1. It would be interesting to examine whether this is a common feature of many or specific proteins that phase separate. We also found that PARP1 autoPARylation reduced the partitioning of shorter DNA fragments in PARP1 condensates as well as the PARP1-dependent compaction of longer DNA. This is in line with a current model that PARylation causes PARP1 to release from DNA (Eustermann et al, 2015; Satoh and Lindahl, 1992) and causes decondensation of nucleosome arrays (Poirier et al, 1982; Strickfaden et al, 2016) and naked DNA (Bell et al, 2021). However, our data adds that the time-dependent enrichment of DNA into the LigIII condensates does not depend on PARylation and demonstrates that the DNA "hand off" can occur between PARP1 and LigIII phases without PARylation-driven removal of PARP1 from DNA or PARP1 condensate dissolution. We also report that PARylation alters PARP1 condensate composition by influencing the partitioning of SSBR proteins. PARylation promotes XRCC1 and FUS enrichment in PARP1 condensates. LigIII, XRCC1, and FUS possess PAR-binding domains (ZnF, BRCT, RGG repeats and RRMs) (Fig. 6B) (Breslin et al, 2015; Polo et al, 2019; Kim et al, 2015; Abbotts and Wilson, 2017), which would enable them to be further enriched upon PARylation. On the other hand, PARylation did not influence the enrichment of Polβ (Fig. EV5D). Further investigation will elucidate how competing affinities for DNA, PAR, and other biomolecules lead to the spectrum of enrichment phenotypes at DNA repair foci. PARylation-dependent enrichment of SSBR proteins serves as an example of compositional control of biomolecular condensates by PARylation and may have implications in other condensates regulated by this post-translational modification, such as stress granules (Leung et al, 2011; Duan et al, 2019; McGurk et al, 2018; Marmor-Kollet et al, 2020) and the nucleolus (Feric et al, 2016; Kim et al, 2019a). Our data indicates that the PARylation-dependent

compositional changes have consequences to condensate properties, including internal dynamics; FUS, which forms highly dynamics condensates (Patel et al, 2015), shows increased PARP1 condensate dynamics upon PAR-driven enrichment.

Our results suggest that the efficiency of DNA ligation is enhanced in the presence of PARP1 condensates. Although ligation efficiency was unchanged in the presence of PARP1 with and without autoPARylation in the dilute solution, it was enhanced in the presence of PARP1 condensates that were induced by autoPARylation (Fig. 5). This suggests that the presence of PARP1 condensates and its activity (PARylation) promote DNA ligation. Because we did not find a condition that led to comparable amounts of PARP1 condensates with or without PARylation, we could not separate these effects. PARP1 condensates and PARylation could enhance ligation in multiple ways. PARP1 condensates organize and concentrate the DNA ligase (LigIII) and its scaffold XRCC1 at their periphery and facilitate DNA 'hand off' from PARP1 to LigIII/XRCC1 phases (Figs. 5 and EV6). PARylation may contribute to the ligation reaction by further enriching XRCC1/LigIII around PARP1 condensates, and increasing the dynamics of PARP1 condensates, which may facilitate accelerated exchange of proteins and DNA. A similar phenomenon was observed in SPOP/DAXX condensates, where increased condensate dynamics correlated with the increased catalytic activity of an E3 ubiquitin ligase, CRL3 (Bouchard et al, 2018). It is worth noting that DNA ligation was enhanced 3-fold in the presence of PARP1/LigIII/XRCC1 condensates, which account for less than 5% of the total reaction mixture. Curiously, PARylation in the presence of HPF1 did not influence the ligation of 601 DNA (Nosella et al, 2024). HPF1 directs PARylation to occur mainly on serine residues, which is a major PARylation site observed in cells (Palazzo et al, 2018) and reduces autoPARylation in vitro. Thus, an intriguing possibility is that autoPARylation, rather than PARylation of LigIII, enhances ligation efficiency. Our data also offer explanations for how PARP1 interacts with LigIII and XRCC1 (Caldecott et al, 1994; Nash et al, 1997; Caldecott et al, 1996; Kubota et al, 1996; Abbotts and Wilson, 2017; Caldecott, 2008) (Fig. 6B), while competing for binding to damaged DNA (Leppard et al, 2003); PARP1 and LigIII/XRCC1 form distinct, yet coexisting phases and damaged DNA prefers the LigIII/XRCC1 phase. The presence of both LigIII and XRCC1 was important for a strong enrichment of DNA in the LigIII/XRCC1 phase, although LigIII seemed to drive this process more (Fig. EV6B).

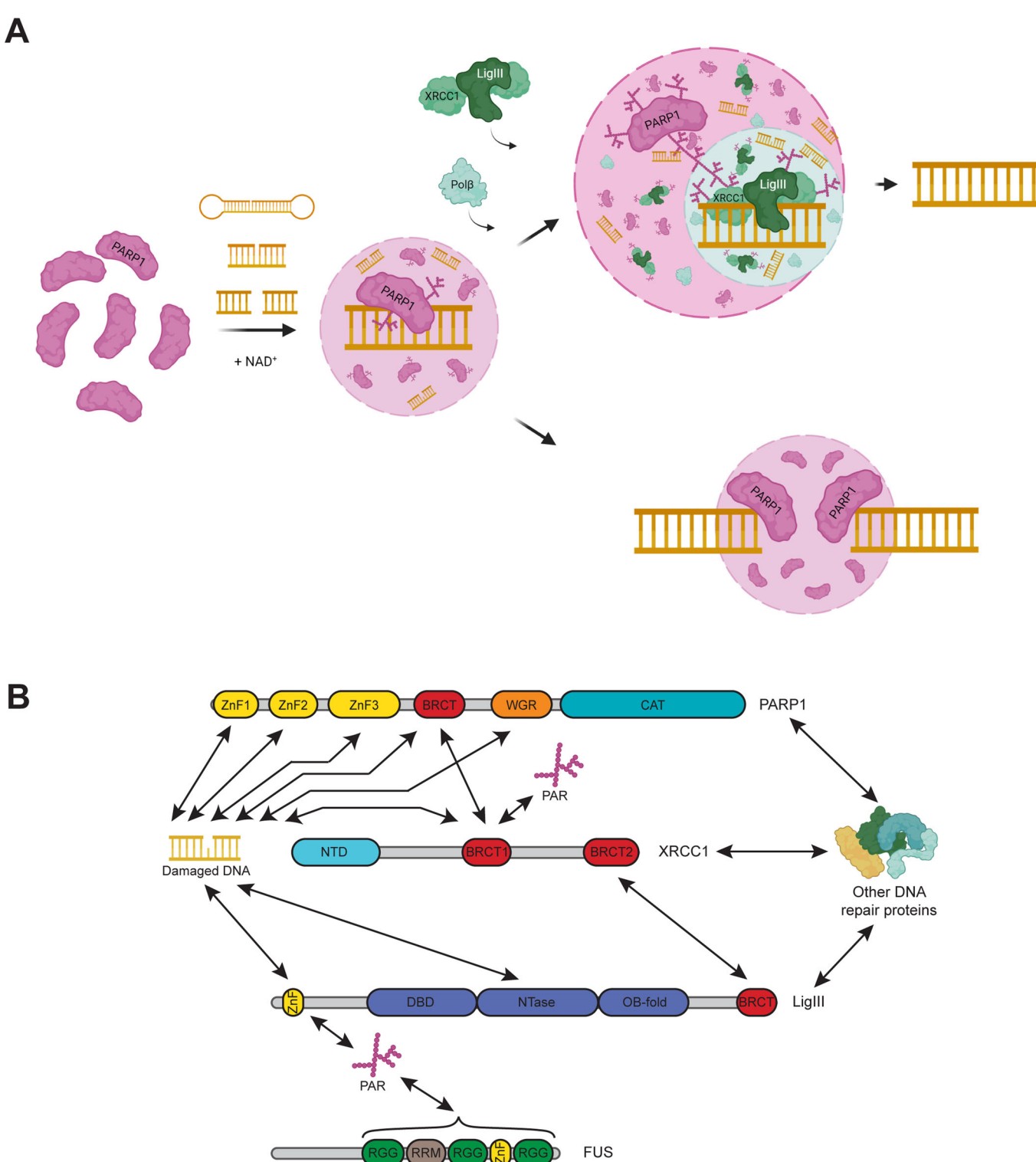

The residence time of PARP1 at DNA damage sites is on the order of minutes in cells, raising the question of how PARP1 condensates disassemble. PAR formation at damage sites is rapidly counteracted by PAR-degrading enzymes, including poly(ADP-ribose) glycohydrolase (PARG) and ADP-ribosyl hydrolase 3 (ARH3) (Barkauskaite et al, 2015). PARG, which preferentially cleaves ADP-ribose from the ends of PAR chains (Barkauskaite et al, 2013), arrives at DNA damage sites with similar kinetics as PARP1 (Aleksandrov et al, 2018). Our data suggests that PARG reverses PARylation-dependent PARP1 condensation, which was in line with findings from a recent study (Chappidi et al, 2024). In addition, PARylation also triggers the recruitment of numerous

**Figure 6. Model of PARP1 condensation in DNA repair.**

(A) When PARP1 binds a DNA lesion (nicked dumbbell, nicked triplex, or double-strand break), it is allosterically activated and PARylates itself, which triggers the formation of a PARP1 condensate (translucent purple circle with a dashed line) that enriches damaged DNA. The recruitment of single-strand break repair proteins XRCC1, LigIII, and Polβ to nicked DNA substrates leads to the formation of multiphase condensates (purple and green translucent circles) and, in combination with PARylation, the enhancement of DNA ligation efficiency. PARP1 condensates can also compact and bridge broken DNA ends together. (B) Schematic showing the reported interactions of PARP1, XRCC1, LigIII, and FUS with each other, PAR, and DNA: PARP1-DNA (Eustermann et al, 2015, 2011; Langelier et al, 2012, 2011; Ali et al, 2012; Rudolph et al, 2021a; Rouleau-Turcotte et al, 2022); PARP-XRCC1 (Masson et al, 1998); XRCC1-LigIII (Taylor et al, 1998; Cuneo et al, 2011); XRCC1-DNA (Polo et al, 2019; Mok et al, 2019); XRCC1-PAR (Breslin et al, 2015; Polo et al, 2019); LigIII-DNA (Taylor et al, 2000; Abdou et al, 2015; Cotner-Gohara et al, 2008, 2010); LigIII-PAR (Leppard et al, 2003); FUS-PAR (Mastrocola et al, 2013; Mamontova et al, 2023). BRCT-PAR and ZnF-PAR interactions depicted in the figure involve multiple protein partners (BRCT-PAR interactions for PARP1, XRCC1, and LigIII; ZnF-PAR interactions for PARP1, LigIII, FUS). The figures were created with BioRender.

repair proteins that also bind DNA at the damage sites (Aleksandrov et al, 2018; Koczor et al, 2021). Since PAR chain length and DNA binding regulate PARP1 condensation, it seems plausible that the combination of PARG activity to degrade PAR and the arrival of other proteins that compete for DNA binding causes PARP1 condensate dissolution over time. Another potential mechanism of PARP1 condensate disassembly is modification of PARP1: for example, cleavage of PARP1 by the cell death protease caspase-3 reduced its condensation. It is tempting to speculate that during apoptosis, caspase-3 activity may limit PARP1 condensate formation. Other post-translational modifications may also contribute to the process.

In summary, our results demonstrate that interactions between PARP1 and damaged DNA leads to the formation of biomolecular condensates that enrich DNA and DNA repair proteins with functional consequences in holding together broken DNA ends and enhancing DNA ligation. Our findings support a model in which PARP1 phase separation and activity form a seed that enhances the condensation of SSBR proteins into sub-compartments that preferentially concentrate repair enzymes with DNA. These findings shed insight into how PARP1 may facilitate SSBR protein recruitment and repair reactions. Future studies in cells would be valuable to understand how these effects at the molecular level contribute to DNA repair in the complex environment of the nucleus.

## Methods

### Reagents and tools table

| Reagent/resource | Reference or source | Identifier or catalog number |
| --- | --- | --- |
| **Experimental models** | | |
| *E. coli* BL21 codon plus (DE3)-RIPL | Agilent Technologies | Cat # 130280 |
| *E. coli* BL21 (DE3) | In-house | N/A |
| **Recombinant DNA** | | |
| pET-SUMO-PARP1 | This study | N/A |
| pET-SUMO-mCherry-PARP1 | This study | N/A |
| pET-SUMO-PARP1-ZnF | This study | N/A |
| pET-SUMO-PARP1 ΔZnF | This study | N/A |
| pET-SUMO-Lig3 | This study | N/A |
| pET-SUMO-XRCC1 | This study | N/A |

| Reagent/resource | Reference or source | Identifier or catalog number |
| --- | --- | --- |
| pET-SUMO-PARG | This study | N/A |
| pET-SUMO-Polβ | This study | N/A |
| pET28a-His-MBP-FUS-GFP | This study | N/A |
| pET-SUMO-Caspase-3 WT | Genscript | N/A |
| pET-SUMO-Caspase-3 C165S | Genscript | N/A |
| pFl Strep Sumo TEV OAS1 | Ando et al, 2019 | N/A |
| pET24a-PARP5a (1093-1327) | Rhine et al, 2022 | N/A |
| **Antibodies** | | |
| Mouse Anti-PAR | Enzo Life Sciences | ALX-804-220-R100 |
| anti-Mouse IgG, HRP | Cell Signaling Technology | 7076 V |
| **Oligonucleotides and other sequence-based reagents** | | |
| DNA substrates and oligos | This study | Appendix Table S2 |
| **Chemicals, enzymes, and other reagents** | | |
| 18:1 (Δ9-Cis) PC (DOPC) | Avanti Polar Lipids | Cat # 850375 P |
| 18:1 Biotinyl Cap PE | Avanti Polar Lipids | Cat # 870273 P |
| 18:1 PEG2000 PE | Avanti Polar Lipids | Cat # 880130 P |
| 2-mercaptoethanol | Sigma-Aldrich | Cat # M3148-250ML |
| ABT-888 | Toronto Research Chemicals | Cat # A112580-10 |
| Alexa Fluor 488 NHS-ester | Thermo Fisher Scientific | Cat # A20000 |
| Benzamidine | BioShop | Cat # BEN601.5 |
| Bovine Serum Albumin | Thermo Fisher Scientific | Cat # BP9706100 |
| Catalase | Sigma-Aldrich | Cat # C100 |
| Chloramphenicol | BioShop | Cat # CLR201.5 |
| Cy3-dATP | Jena Bioscience | Cat # NU-835-CY3 |
| DNase | Roche Diagnostics | Cat # 4536282001 |
| DTT | Goldbio | Cat # DTT50 |
| EDTA | Wisent Bioproducts | Cat # 625-060-CG |

| Reagent/resource | Reference or source | Identifier or catalog number |
|---|---|---|
| Gibson Assembly Master Mix | New England Biolabs | Cat # E2611S |
| Glucose Oxidase | Sigma-Aldrich | Cat # G2133 |
| Glycerol | BioShop | Cat # GLY002.4 |
| Glycine | BioShop | Cat # GLN001.1 |
| HEPES | BioShop | Cat # HEP005.100 |
| Histones | Sigma-Aldrich | Cat # H9250 |
| Imidazole | BioShop | Cat # IMD508.100 |
| isopropyl β-D-1-thiogalactopyranoside (IPTG) | BioShop | Cat # IPT001.5 |
| Kanamycin | BioShop | Cat # KAN201.5 |
| Kapa HiFi Hotstart ReadyMix | Roche Diagnostics | Cat # 07958927001 |
| Lambda DNA | New England Biolabs | Cat # N3011S |
| LB Broth (Miller) | BioShop | Cat # LBL407.5 |
| Luminata Crescendo Western HRP Substrate | Sigma-Aldrich | Cat # WBLUR0500 |
| Lysozyme | BioShop | Cat # LYS702.5 |
| Magnesium chloride | BioShop | Cat # MAN222.100 |
| Methanol | Caledon Laboratories | Cat # 6701-7-40 |
| Nicotinamide adenine dinucleotide (NAD⁺) | New England Biolabs | Cat # B9007S |
| Ni-NTA Agarose | Qiagen | Cat # 30210 |
| Nt.BspQI | New England Biolabs | Cat # R0644S |
| PMSF | BioShop | Cat # PMS123.25 |
| Poly(I:C) RNA | Invivogen | Cat # Tlrl-picw |
| Potassium chloride | BioShop | Cat # POC888.1 |
| QuickChange Site-Directed Mutagenesis Kit | Agilent | Cat # 200515 |
| RNase A | Thermo Fisher Scientific | Cat # EN0531 |
| SfoI | New England Biolabs | Cat # R0606S |
| Skim milk powder | BioShop | Cat # SKI400.1 |
| Sodium chloride | BioShop | Cat # SOD002.5 |
| Streptavidin | Thermo Fisher Scientific | Cat # 434302 |
| SYTOX Orange Nucleic Acid Stain | Thermo Fisher Scientific | Cat # S11368 |
| T4 DNA Ligase | New England Biolabs | Cat # M0202S |
| Tris | BioShop | Cat # TRS001.1 |
| tris(2-carboxyethyl) phosphine (TCEP) | BioShop | Cat # TCE101.1 |
| Tween-20 | BioShop | Cat # TWN510.500 |

| Reagent/resource | Reference or source | Identifier or catalog number |
|---|---|---|
| Zinc chloride | BioShop | Cat # ZNC233.100 |
| **Software** | | |
| Fiji | https://fiji.sc/ | |
| Matlab | https://www.mathworks.com/products/matlab.html | |
| GraphPad Prism | https://www.graphpad.com/ | |
| Biorender | https://www.biorender.com/ | |
| ImageStudio | https://www.licor.com/bio/image-studio/ | |
| Adobe Illustrator | https://www.adobe.com/ | |
| **Other** | | |

## Cloning and construct generation

Gibson assembly (New England Biolabs) was used to generate all constructs. cDNAs encoding full-length, wild-type *Homo sapiens PARP1*, *PARG*, *LIG3* (splice isoform alpha, containing C-terminal BRCT domain), and *XRCC1* were obtained from the NIH Mammalian Gene Collection (MGC). DNA fragments containing the full-length coding sequences of each protein (PARP1 1-1014, PARG 1-976, XRCC1 1-633, LIG3 43-1009, excluding mitochondrial targeting sequence) were amplified by polymerase chain reaction (PCR) using Kapa HiFi Hotstart ReadyMix (Roche) and inserted into a pET-SUMO expression vector containing an N-terminal hexahistidine-SUMO tag (Invitrogen). PARP1 truncations (PARP1-ZnF, residues 1–378; PARP1-ΔZnF, residues 379–1014) were generated by PCR from PARP1 cDNA and inserted into a pET-SUMO expression vector. The DNA for full-length, wild-type human caspase-3 was synthesized by GenScript (Piscataway, NJ, USA), codon-optimized for expression in *E. coli*, and subsequently cloned into a pET-SUMO expression vector with an additional C-terminal hexahistidine tag cloned in using QuikChange (Agilent).

## Protein expression and purification

### PARP1, PARP1-ZnF, PARP1-ΔZnF, PARG, XRCC1, and Polβ

pET-SUMO expression vectors containing His-SUMO-tagged PARP1 constructs were transformed into *E. coli* BL21 (DE3) RIPL cells and grown in LB media with kanamycin and chloramphenicol. After overnight growth for 16 hr at 37 °C and 200 rpm, overnight cultures were used to inoculate large-scale cultures at a starting OD of ~0.2. Construct expression was induced at an OD of ~1.0 with 0.2 mM isopropyl β-D-1-thiogalactopyranoside (IPTG) and 0.1 mM zinc chloride and grown at 16 °C for 16–20 h at 200 rpm. Cells were harvested by centrifugation and lysed in a buffer containing 25 mM HEPES pH 7.5, 250 mM NaCl, 20 mM imidazole, 10% v/v glycerol, and 5 mM 2-mercaptoethanol with DNase and lysozyme by sonication (30% amplitude, 2 s pulses/50% duty for 9 min total). Lysed cells were sedimented by centrifugation at 20,000 RCF at 4 °C for 30 min, and the lysate was subsequently loaded onto pre-

equilibrated Ni-NTA agarose resin (Qiagen) in a gravity column at 4 °C. The resin was washed with 5-column volumes of lysis buffer, and the protein of interest was eluted using a step-wise elution in lysis buffer containing 50 mM, 100 mM, and 250 mM imidazole. Eluted protein was subjected to His-SUMO tag cleavage with ULP1 protease (purified in-house) in dialysis buffer containing 25 mM HEPES pH 7.5, 250 mM NaCl, 5 mM 2-mercaptoethanol overnight at 4 °C. Cleaved protein was subsequently separated from the His-SUMO tag and protease by re-loading onto Ni-NTA resin and collecting the flow-through in lysis buffer. The flow-through was then concentrated in an Amicon centrifugal unit (EMD-Millipore) with the appropriate molecular weight cutoff and purified on an AKTA Go FPLC system (Cytiva) using a Superdex 200 16/600 HiLoad column (Cytiva) pre-equilibrated in 25 mM Tris pH 7.5, 1 M NaCl, 0.5 mM EDTA, 2 mM 2-mercaptoethanol at 4 °C. Fractions containing the protein of interest were pooled, and purified on an AKTA Go FPLC system (Cytiva) using a pre-packed 4-mL source Q anion exchange resin (Cytiva) pre-equilibrated in 25 mM HEPES pH 7.5, 100 mM NaCl, 0.5 mM EDTA, and 2 mM 2-mercaptoethanol at 4 °C. Fractions containing protein of interest were pooled and dialyzed into 25 mM Tris pH 7.5, 100 mM NaCl, 0.5 mM EDTA, 5 mM 2-mercaptoethanol and concentrated. Purity was assessed by SDS-PAGE. PARG, XRCC1, and Polβ were purified using the same protocol, but bacterial cultures were not supplemented with $ZnCl_2$.

### LigIII

pET-SUMO expression vector with His-SUMO-tagged LigIII-alpha was transformed into *E. coli* BL21 (DE3) RIPL cells and grown in LB with kanamycin and chloramphenicol. After the growth of overnight cultures for 16 h at 37 °C and 200 rpm, overnight cultures were used to inoculate large-scale cultures at a starting OD of ~0.2. Construct expression was induced at an OD of ~0.6 with 0.3 mM isopropyl β-D-1-thiogalactopyranoside (IPTG) and grown at 16 °C for 16–20 h at 200 rpm. Cells were harvested by centrifugation and lysed in buffer containing 50 mM Tris pH 7.5, 300 mM NaCl, 10% glycerol, 20 mM imidazole, 1 mM benzamidine, 2 mM 2-mercaptoethanol with DNase and lysozyme by sonication (30% amplitude, 2 s pulses/50% duty for 9 min total). Lysed cells were sedimented by centrifugation at 20,000 RCF at 4 °C for 30 min, and the lysate was subsequently loaded onto pre-equilibrated Ni-NTA agarose resin (Qiagen) at 4 °C. The resin was washed with 5-column volumes of lysis buffer, and the protein of interest was eluted using a step-wise elution with lysis buffer containing 100, 200, and 600 mM imidazole. Eluted protein was subjected to His-SUMO tag cleavage with ULP1 protease in dialysis buffer containing 50 mM Tris pH 7.5, 300 mM NaCl, and 2 mM 2-mercaptoethanol overnight at 4 °C. Cleaved protein was subsequently separated from the His-SUMO tag and protease by re-loading onto Ni-NTA resin and collecting the flow-through in lysis buffer. The flow-through was then loaded onto a HiTrap Blue HP column (Cytiva) in 50 mM Tris pH 7.5, 300 mM NaCl, 10% glycerol, 1 mM benzamidine, and 2 mM 2-mercaptoethanol. The column was washed with the same buffer and eluted with a step-wise salt gradient (500 mM NaCl, 1 M NaCl, 2 M NaCl). The elution fractions were then pooled and concentrated in an Amicon centrifugal unit (EMD-Millipore) with a 100 kDa MWCO membrane and purified on an AKTA Go FPLC system (Cytiva) using a Superdex 200 pg 16/600 HiLoad column (Cytiva) pre-equilibrated

in 25 mM Tris pH 7.5, 500 mM NaCl, 0.5 mM EDTA, 1 mM tris(2-carboxyethyl)phosphine (TCEP) at 4 °C. Fractions containing the protein of interest were concentrated and purity was assessed by SDS-PAGE.

### Caspase-3

Full-length caspase-3 was expressed and purified as outlined in previous literature (Stennicke and Salvesen, 1999), with additional details included below for clarity. Caspase-3 initially exists in its zymogenic form (procaspase-3) and requires proteolytic cleavage for complete activation. Procaspase-3 is activated via cleavage at D9 and D28 in the prodomain, which subsequently dissociates, and D175 in the intersubunit linker to yield the p20 and p10 subunits, which then form a dimer of dimers (p202p102) (Riedl and Shi, 2004). Because procaspase-3 is activated during overexpression in *E. coli*, any affinity tag that precedes the prodomain will dissociate from mature caspase-3 upon activation. For this reason, a C-terminal hexahistidine tag was cloned into the caspase-3 expression vector. The caspase-3 expression vector was transformed into *E. coli* BL21(DE3) competent cells, which were grown in LB medium at 37 °C with 30 µg/mL kanamycin and shaking at 220 rpm. When the optical density reached 0.6 units, the temperature was lowered to 16 °C, and isopropyl β-D-1- thiogalactopyranoside (IPTG) was added to a final concentration of 0.2 mM. Caspase-3 expression proceeded for 18 h at 16 °C and the cells were harvested by centrifugation and stored at −80 °C until use. The cell pellet was resuspended in 25 mM HEPES, 150 mM NaCl, 0.5 mM 2-mercaptoethanol, and 30 mM imidazole at pH 8.0. Following lysis by sonication and clarification of the lysed material by centrifugation, the supernatant was loaded onto a 5 mL HisTrap HP column (Cytiva) in the same buffer listed above. Caspase-3 was eluted with 25 mM HEPES, 150 mM NaCl, 0.5 mM 2-mercaptoethanol, and 300 mM imidazole at pH 8.0. The eluted material was concentrated in an Amicon Ultra-15 3 kDa MWCO concentrator and loaded onto a HiLoad 16/600 Superdex 200 pg column (Cytiva) in buffer containing 25 mM HEPES, 50 mM NaCl, 5 mM tris(2- carboxyethyl)phosphine (TCEP) at pH 8.0. Fractions that contained caspase-3 were pooled, concentrated in an Amicon Ultra-15 3 kDa MWCO concentrator, and stored at −80 °C until use.

### FUS-GFP

Full-length FUS tagged with a C-terminal GFP tag and an N-terminal His-MBP tag was overexpressed in BL21 (DE3) RIPL *E. coli* cells at 16 °C for 16 h after induction with 500 µM IPTG. Cells were harvested and lysed by sonication in lysis buffer (50 mM Tris-HCl, 500 mM NaCl, pH 8.0, 10 mM imidazole, 4 mM β-mercaptoethanol, 1 mM PMSF, and 0.1 mg/mL RNase A). The cell pellet was removed after centrifugation (16,000 rpm, 4 °C, 1 h), and the supernatant was loaded into a Ni-NTA agarose column (Qiagen). Protein was eluted in an elution buffer (50 mM Tris-HCl, 500 mM NaCl, pH 8.0, 100 mM imidazole, and 4 mM β-mercaptoethanol). The His-MBP tag was cleaved using a His-tagged TEV protease during dialysis against FUS dialysis buffer (50 mM Tris-HCl pH 7.4, 1 M KCl, 10% glycerol, 4 mM β-mercaptoethanol) overnight at 4 °C. The mixture was loaded onto a Ni-NTA agarose column, and FUS-GFP was collected in the flow-through fractions. The protein was further purified by gel filtration chromatography (HiLoad Superdex 200 pg 16/600; Cytiva),

concentrated, and equilibrated with storage buffer (50 mM Tris-HCl, 500 mM KCl, 2 mM DTT, and 10% glycerol). Peak fractions were pooled and flash-frozen in liquid nitrogen and stored at −80 °C.

### Fluorescent protein labeling

About 1–10 µmol of protein of interest was dialyzed into 25 mM HEPES pH 7.5, 250 mM NaCl, 0.5 mM EDTA, 0.5 mM tris(2-carboxyethyl)phosphine (TCEP) overnight using 3.5 kDa MWCO Slide-A-Lyzer dialysis cassettes (Thermo Fisher Scientific). The protein was combined with a 5× molar excess of Alexa Fluor 488 NHS-ester dye (Thermo Fisher Scientific) dissolved in DMSO and incubated overnight at 4 °C in the dark. The reaction was quenched with 5× excess volume of 25 mM Tris pH 7.5, 250 mM NaCl, 0.5 mM EDTA, 5 mM 2-mercaptoethanol and desalted into the same buffer using a 5 mL HiTrap desalting column (Cytiva) using an AKTA FPLC system (Cytiva) at 4 °C. Fractions containing labeled protein were visualized by Alexa Fluor 488 fluorescence on a BioRad ChemiDoc MP system with Alexa Fluor 488 excitation/emission settings. Fractions containing labeled protein but no free dye were pooled and concentrated to the desired concentration and stored at −80 °C.

### In vitro PARylation assay

PARP1 at the indicated concentrations was combined with 0.3 µM nicked dumbbell DNA (Appendix Table S2) in PARylation Buffer (20 mM Tris pH 7.5, 50 mM NaCl, 7.5 mM MgCl₂, and 1 mM DTT). About 0.5 mM NAD⁺ was added to initiate the reaction, and the reaction progression and/or phase separation were analyzed by SDS-PAGE, or microscopy at indicated timepoints. For experiments involving additional proteins (i.e., XRCC1-LigIII), these components were added to the reactions at the indicated concentrations prior to reaction initiation (unless indicated otherwise). For experiments involving labeled DNA oligonucleotides, Cy5-labeled triplex DNA (sequence found in Appendix Table S2) was added to the reaction in lieu of the dumbbell DNA at the indicated concentrations prior to reaction initiation. For experiments involving ABT-888 (Toronto Research Chemicals Inc), ABT-888 was added to reactions immediately prior to reaction initiation.

### Western blot

Proteins were resolved onto a 4–20% Mini-PROTEAN TGX Precast gels (BioRad) and then transferred to a 0.2 µM nitrocellulose membrane (Biorad) in cold transfer buffer (25 mM tris base, 190 mM glycine, and 20% methanol) for 1 h at 90 V (4 °C). The nitrocellulose membrane was then blocked with blocking buffer (5% w/v skim milk powder, 0.05% v/v tween-20 in tris-buffered saline (TBST)) before its incubation with primary antibody (anti-PAR) overnight at 4 °C. The membrane was then incubated in horseradish peroxidase (HRP)-linked secondary antibody (anti-mouse IgG) for 1 h at room temperature and developed on a Biorad ChemiDoc MP imaging system using Luminata Crescendo Western HRP Substrate (Sigma, WBLUR0500).

### Expression and purification of PARP5a (1093-1327)

PARP5a catalytic domain was expressed and purified as described (Rhine et al, 2022).

### Expression and purification of OAS1

OAS1 was expressed and purified as described (Ando et al, 2019).

### PAR synthesis and fractionation to PAR16

PAR was prepared essentially as described (Ando et al, 2019). PAR was synthesized enzymatically by PARP5 catalytic domain (0.1 mg/mL) with histones (2 mg/mL, Sigma #H9250) and NAD⁺ (20 mM) in 50 mM HEPES pH 7, 10 mM MgCl₂, 1 mM DTT, 0.02% v/v NP-40 for 2 h at ambient temperature. PARylated proteins were precipitated with an equal volume of 20% w/v trichloroacetic acid (10% w/v final) and pelleted with 20,000 × g for 30 min at 4 °C. Pellets were washed with 70% ethanol, then PAR was cleaved from the proteins with 0.5 M KOH, 50 mM EDTA for 1 h at 60 °C. Sodium acetate pH 5.2 was added to a final concentration of 0.3 M, then ethanol was added to a final concentration of 70% v/v. PAR was precipitated for 1 h at −80 °C, then pelleted with 20,000 × g for 30 min at 4 °C. PAR pellets were washed with 70% ethanol, dried at 37 °C for 10 min, then resuspended in water. The mixture of PAR lengths was fractionated to PAR16 essentially as described (Ando et al, 2019). A preparative DNAPac PA100 column (22 × 250 mm) fractionated PAR into homogeneous polymers. PAR (up to 20 µmol ADPr) was loaded onto the column equilibrated with Dionex buffer A (25 mM Tris-HCl pH 9.0), and the concentration of Dionex buffer B (25 mM Tris-HCl pH 9.0 and 1 M NaCl) in a 120-min method was set to elute as follows: 0 min (0% B), 6 min (0% B), 10 min (30% B), 60 min (40% B), 78 min (50% B), 108 min (56% B), 112 min (100% B), 114 min (100% B), 115 min (0% B), 120 min (0% B). Fractions containing PAR16 were combined, concentrated, and desalted into water with Amicon centrifugal filters (3k MWCO). [PAR16] was measured with a NanoDrop OneC using the equation: $[PAR16] = n \times A_{260}/13,500 \text{ M}^{-1} \text{ cm}^{-1}$, where $n$ is the number of adenines, in this case 16.

### PAR chain length analysis of a PARP1 autoPARylation reaction

PAR from a PARP1 autoPARylation reaction was isolated, fluorescently labeled, and analyzed with PAGE essentially as described (Ando et al, 2019). AutoPARylated mCherry-PARP1 was acid precipitated, and PAR was isolated with base and ethanol precipitation as described above for PAR16 synthesis. The mixture of PAR lengths (~250 µM ADPr) was labeled with Cy3-dATP (10 µM, Jena Bioscience #NU-835-CY3), poly(I:C) RNA (50 µg/mL, Invivogen #tlrl-picw), and OAS1 (1 µM) in 25 mM HEPES pH 7.5, 20 mM MgCl₂, 2.5 mM DTT at 37 °C for 2 h. Labeling reactions were diluted with Formamide-EDTA loading buffer and separated with 12% urea-PAGE (National Diagnostics # EC-833) in an adjustable slab gel apparatus (VWR #CBASG-400) equipped with 28 cm plates. Cy3 signal was measured with a Licor-M. The image was exported and annotated with ImageStudio and Adobe Illustrator.

### SDS-PAGE

Protein samples were quenched in 4x SDS-PAGE sample buffer (Biorad) to a final concentration of 1x. Samples were loaded onto 4–20% Mini-PROTEAN TGX precast gradient gels (Biorad) and resolved for 35 min at 200 V, and then stained with Coomassie blue (prepared in-house) and destained overnight. Gels were imaged on a BioRad ChemiDoc MP system.

### DIC microscopy of PARP1 mutant condensates

PARylation reactions were set up as described above in PARylation Buffer (20 mM Tris pH 7.5, 50 mM NaCl, 7.5 mM MgCl₂, and 1 mM DTT) and transferred onto 35-mm-diameter glass-bottom dishes (MatTek) and sealed with a glass coverslip to limit

evaporation. Images were acquired on a Nikon Ti-2E microscope with an Andor Dragonfly 200 confocal spinning disk unit using a 40x oil immersion objective (DIC channel) and a Zyla sCMOS camera.

### Fluorescence microscopy

PARylation reactions were set up as described above in PARylation Buffer (20 mM Tris pH 7.5, 50 mM NaCl, 7.5 mM MgCl$_2$, and 1 mM DTT), transferred onto 35-mm-diameter glass-bottom dishes (MatTek), and sealed with a glass coverslip to limit evaporation. Images were acquired on a Nikon Ti-2E microscope with an Andor Dragonfly 200 confocal unit using a 60x oil immersion objective and 2048 × 2048-pixel resolution. Fluorescence was detected using a Zyla sCMOS camera after excitation with 488, 561, and 637 nm lasers. Images were processed and analyzed using Fiji. Photobleaching was performed using the LASX FRAP module on the Leica SP8 microscope. Circular regions of interest (ROI) of 1 µm × 1 µm size were positioned in the center of droplets (PARP1). Photobleaching was performed with the 552 nm laser at 100% laser power for three repetitions on zoom-in mode, and images were taken at 5 s intervals. Fluorescence recovery over time was normalized to the intensity of images taken pre-bleach and plotted in GraphPad Prism. For condensate area analysis, the area covered by condensates was determined by setting an intensity threshold and dividing by the area of the whole image. For protein enrichment analysis, the average fluorescence intensity within condensates was divided by the average fluorescence intensity in the dilute phase. Dumbbell DNA, Cy5-Triplex, and 20, 40, and 60-nt DNA oligonucleotides were ordered from IDT. To generate the nicked pUC19 DNA substrate, the pUC19 plasmid was digested with Nt.BspQI (NEB, R0644S), purified (Thermo, K0701) and analyzed on an agarose gel to confirm complete digestion.

### Single-molecule fluorescence microscopy

Single-molecule fluorescent images were collected using a customized prism TIRF microscope (Kim et al, 2019c; Soniat et al, 2017). An inverted Nikon Ti-E microscope system was equipped with a motorized stage (Prio ProScan II H117). The flowcell was illuminated with a 488 nm laser (Coherent Sapphire) and a 637 nm laser (Coherent OBIS) through a quartz prim (Tower Optical Co.). For imaging SYTOX Orange-stained DNA, the 488 nm laser power was adjusted to deliver low power (4 mW) at the front face of the prism using a neutral density filter set (Thorlabs). For short DNA-ATTO647N capture experiments, the 637 nm laser power was adjusted to 10 mW. Images were recorded using two electron-multiplying charge-coupled device (EMCCD) cameras (Andor iXon DU897). Unless indicated, DNA was extended via continuous buffer flow (0.15 mL min$^{-1}$) in the imaging buffer (40 mM Tris-HCl pH 7.5, 25 mM NaCl, 2 mM MgCl$_2$, 0.2 mg mL$^{-1}$ BSA, 1 mM DTT, 100 nM SYTOX Orange) supplemented with an oxygen scavenging system (3% D-glucose (w/v), 1 mM Trolox, 1500 units catalase, 250 units glucose oxidase; all from Sigma-Aldrich). NAD$^+$ (Spectrum Chemical) was added in the imaging buffer at the indicated concentration. NIS-Elements software (Nikon) was used to collect the images at a 0.28–25 s frame rate with 80 ms exposure time. All images were exported as uncompressed TIFF stacks for further analysis in FIJI (NIH) and MATLAB (The MathWorks).

Flowcells were assembled with a 4-mm-wide, 100-µm-high flow channel between a glass coverslip (VWR) and a custom-made quartz microscope slide using two-sided tape (3 M). DNA curtains were prepared with 40 µL of liposome stock solution (97.7% DOPC (Avanti #850375 P), 2.0% DOPE- mPEG2k (Avanti #880130 P), and 0.3% DOPE-biotin (Avanti #870273 P) in 960 µL Lipids Buffer (10 mM Tris-HCl, pH 8.0, 100 mM NaCl) incubated in the flowcell for 30 min. Then, the flowcell was washed with BSA Buffer (40 mM Tris-HCl, pH 8.0, 2 mM MgCl$_2$, 1 mM DTT, 0.2 mg mL$^{-1}$ BSA) and incubated for 10 min. Streptavidin (0.1 mg mL$^{-1}$ diluted in BSA buffer) was injected into the flowcell for another 10 min. Finally, ~12.5 ng µL$^{-1}$ of DNA substrate was introduced into the flowcell. Subsequently, 100 µL of 1 unit mL$^{-1}$ SfoI restriction enzyme (NEB) was injected to generate blunt-end DNA molecules.

### DNA substrates for single-molecule imaging

To prepare DNA substrates for microscopy, 125 µg of λ-phage DNA was mixed with two oligos (2 µM oligo Lab07 (/5Phos/AGG TCG CCG CCC/3BioTEG) and 2 µM oligo Lab06 (/5Phos/GGG CGG CGA CCT/3BioTEG) in 1× T4 DNA ligase reaction buffer (NEB B0202S) and heated to 70 °C for 15 min followed by gradual cooling to 15 °C for 2 h. One oligo will be annealed with the overhang located at the left cohesive end of DNA, and the other oligo will be annealed with the overhang at the right cohesive end. After the oligomer hybridization, 2 µL of T4 DNA ligase (NEB #M0202S) was added to the mixture and incubated overnight at room temperature to seal nicks on the DNA. The ligase was inactivated with 2 M NaCl, and the reaction was resolved over a custom-packed S-1000 gel filtration column (GE) to remove excess oligonucleotides and proteins.

For the preparation of 60 bp short DNA labeled with ATTO647N, Oligo1 (/5BioTEG/ACGAAGTCTTATGGCAAAA-CCGATGGACTATGTTTCGGGTAGCACCAGAAGTCTATAACA) and ATTO647N-tagged Oligo2 (5TGTTATAGACTTCTGGTGC-TACCCGAAACATAGTCCATCGGTTTTGCCATAAGACTTCGT/3ATTO647N/) were purchased from IDT and annealed by combining 20 µM Oligo1 with 20 µM Oligo2. The annealing process involved heating to 75 °C for 10 min, followed by a gradual cooling to 22 °C over a period of 1 h in a thermal cycler.

### Ligation assay

PARylation reactions were set up as described above in PARylation Buffer (20 mM Tris pH 7.5, 50 mM NaCl, 7.5 mM MgCl$_2$, and 1 mM DTT) before Cy5-labeled triplex DNA and ATP added to allow ligation at 30 °C for the indicated period of time before quenching the reaction with 2x quenching buffer (1x TBE, 12% Ficoll 400, 7 M urea) to a final concentration of 1x. Reactions were then incubated at 95 °C for 5 min to denature the DNA. Samples were loaded on a denaturing 7 M urea gel and resolved for 30 min 180 V. Gels were imaged using the Cy5 filter on a BioRad ChemiDoc MP system. Ligation efficiency analysis was performed on Fiji by obtaining the ratio of the ligated and unligated products.

## Data availability

This study includes no data deposited in external repositories. The source data of this paper are linked to the manuscript and stored in the Biostudies database.

The source data of this paper are collected in the following database record: biostudies:S-SCDT-10_1038-S44319-024-00285-5.

## Peer review information

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

## Acknowledgements

We would like to thank members of the Forman-Kay, H. Lee, and Ditlev labs (Biochemistry, University of Toronto) for stimulating discussions and critical reading of the manuscript. We thank Fiona Cui for their assistance with bioinformatic analysis. CCS and MLN are funded by a Canada Graduate Scholarship-Doctoral from the Natural Sciences and Engineering Research Council (NSERC). MT is funded by an NSERC Vanier Canada Graduate Scholarship. HZ and IJF are funded by NCI grant 5P01CA092584 and the Welch Foundation (F-1808). AKLL and MD are funded by NIH R01GM04135 and T32GM080189. TRA is funded by a Banting Postdoctoral Fellowship from the Canadian Institutes of Health Research (CIHR). This work was funded by NSERC RGPIN-2019-070 to HOL, CIHR FDN-148375 to JDF-K, and the Canada Research Chair program to both JDF-K and HOL.

## Author contributions

**Christopher Chin Sang**: Conceptualization; Formal analysis; Validation; Investigation; Visualization; Methodology; Writing—original draft; Writing—review and editing. **Gaelen Moore**: Conceptualization; Formal analysis; Validation; Investigation; Visualization; Methodology; Writing—original draft; Writing—review and editing. **Maria Tereshchenko**: Conceptualization; Formal analysis; Validation; Investigation; Visualization; Methodology; Writing—original draft. **Hongshan Zhang**: Conceptualization; Formal analysis; Validation; Investigation; Visualization; Methodology; Writing—original draft; Writing—review and editing. **Michael L Nosella**: Conceptualization; Formal analysis; Validation; Investigation; Visualization; Methodology; Writing—original draft. **Morgan Dasovich**: Conceptualization; Formal analysis; Validation; Investigation; Visualization; Methodology; Writing—original draft; Writing—review and editing. **T Reid Alderson**: Conceptualization; Resources. **Anthony K L Leung**: Conceptualization; Formal analysis; Supervision; Funding acquisition; Investigation; Methodology; Writing—original draft; Writing—review and editing. **Ilya J Finkelstein**: Conceptualization; Formal analysis; Supervision; Funding acquisition; Validation; Investigation; Visualization; Methodology; Writing—original draft; Writing—review and editing. **Julie D Forman-Kay**: Conceptualization; Formal analysis; Supervision; Funding acquisition; Validation; Investigation; Visualization; Methodology; Writing—original draft; Writing—review and editing. **Hyun O Lee**: Conceptualization; Resources; Formal analysis; Supervision; Funding acquisition; Validation; Investigation; Visualization; Methodology; Writing—original draft; Project administration; Writing—review and editing.

Source data underlying figure panels in this paper may have individual authorship assigned. Where available, figure panel/source data authorship is listed in the following database record: biostudies:S-SCDT-10_1038-S44319-024-00285-5.

## Disclosure and competing interests statement

The authors declare no competing interests.

# Expanded View Figures

**Figure EV1. PARP1 forms condensates in a DNA-dependent manner.**

(A) IUPRED3 disordered region plots for PARP1 (https://iupred.elte.hu/plot). (B) Image of Coomassie Blue stained SDS-PAGE of purified recombinant mCherry-PARP1. (C) Fluorescence micrographs of 4 µM mCherry-PARP1 with or without the indicated concentration of dumbbell DNA. (D) Fluorescence micrographs of 4 µM mCherry-PARP1 with the indicated lengths and concentrations of double-stranded DNA (dsDNA). (E) Fluorescence micrographs of 4 µM mCherry-PARP1 with the indicated concentrations of pUC19 plasmid DNA with or without a single nick. (F) Fluorescence micrographs of 4 µM mCherry-PARP1 and 4 µM triplex DNA condensates before and after addition of 100 mM NaCl to dissolve pre-existing mCherry-PARP1 condensates. (G) Fluorescence micrographs of 4 µM mCherry-PARP1 with or without 4 µM triplex DNA in the presence or absence of 10% 1,6-hexanediol. (H) Phase diagram of condensates formed by recombinant mCherry-PARP1 at indicated concentrations with or without single-stranded DNA. (I) Representative fluorescence micrographs of mCherry-PARP1 condensates diagramed in (H). All figures represent findings from at least three biological replicates.

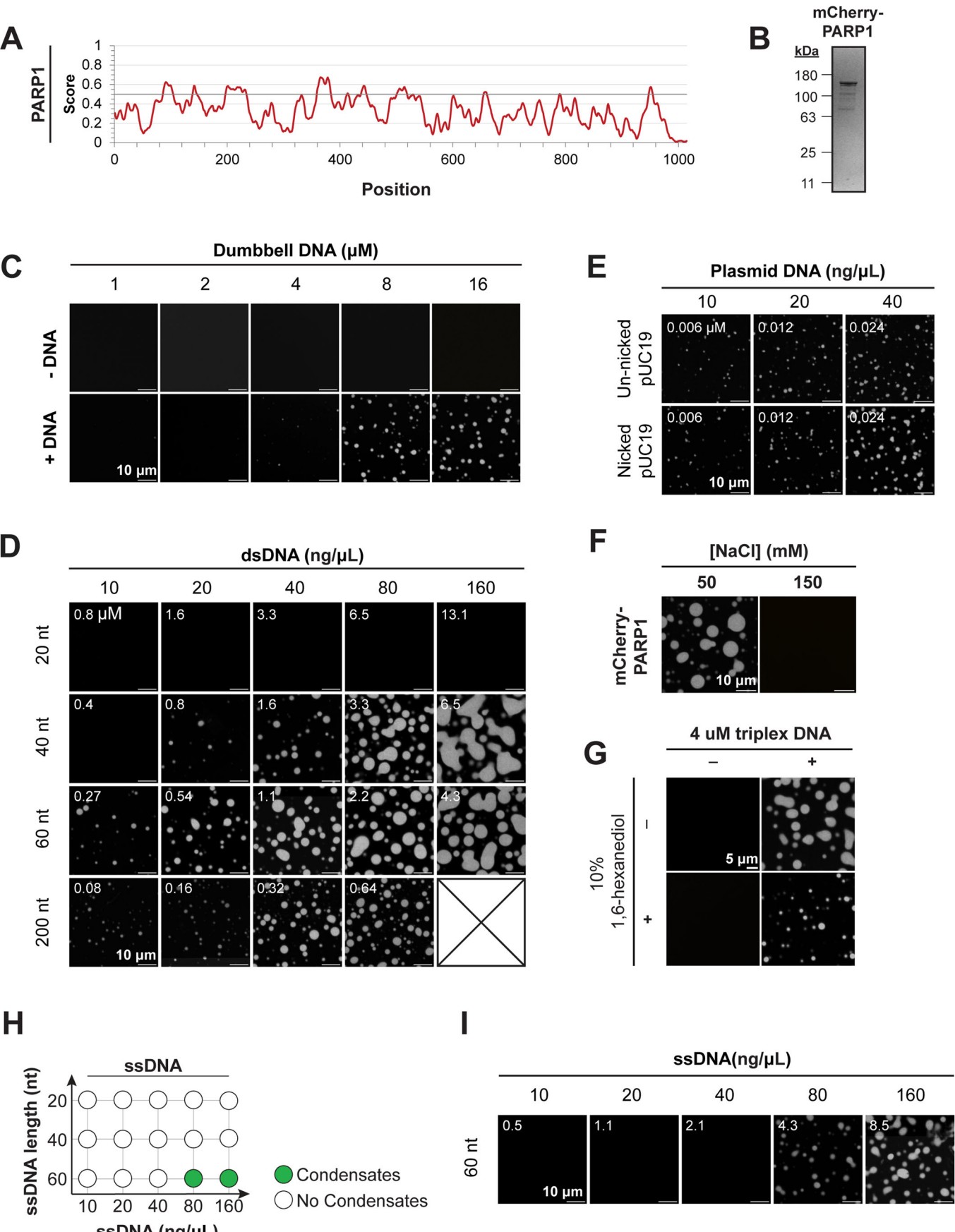

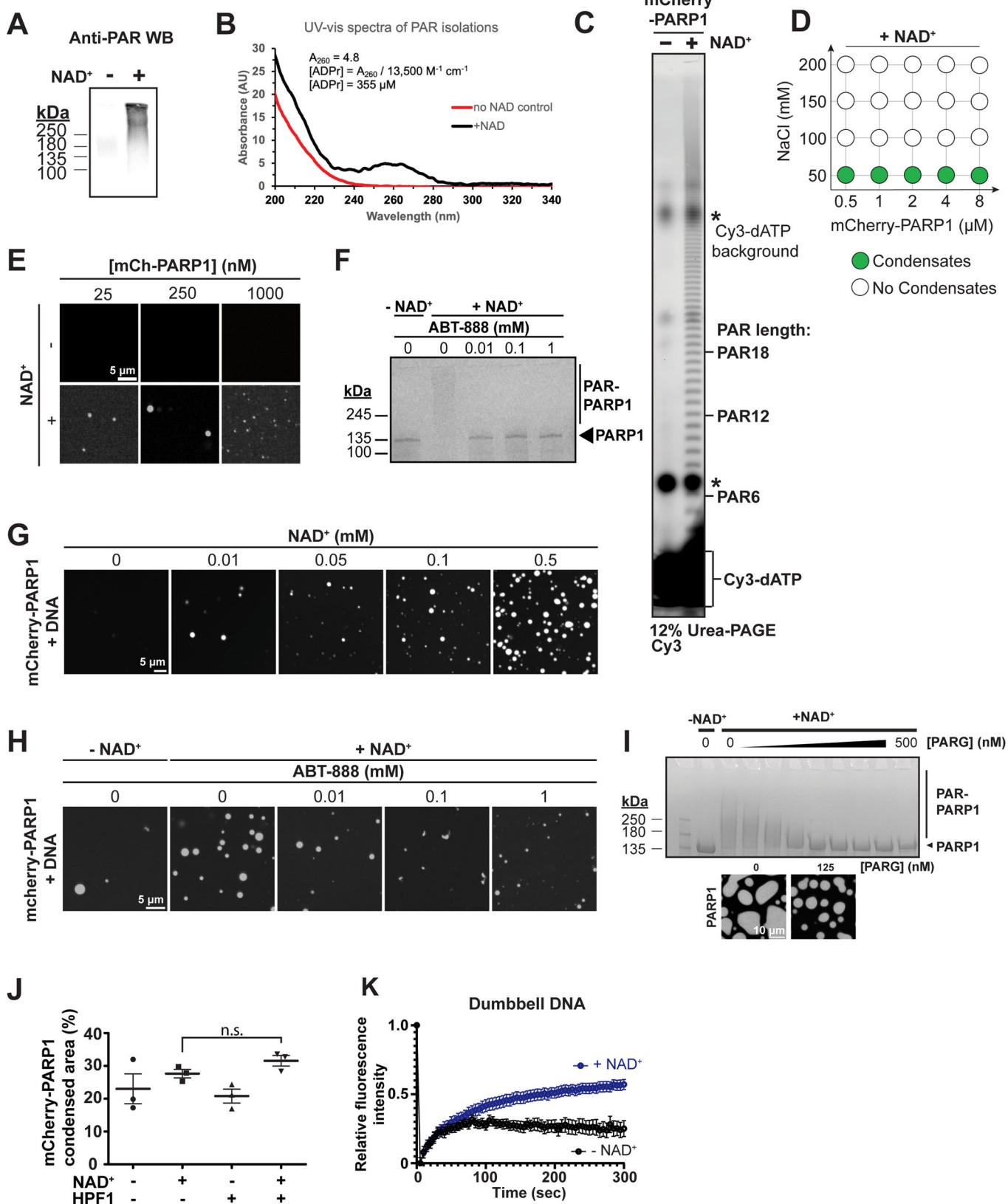

◄  **Figure EV2.  AutoPARylation enhances the formation and internal dynamics of PARP1 condensates.**

(A) Anti-PAR Western blot depicting autoPARylation of PARP1 from reactions of $1\,\mu M$ untagged PARP1 incubated with $0.3\,\mu M$ dumbbell DNA for 15 min with and without $0.5\,mM$ $NAD^+$. (B) UV-Vis spectra of mCherry-PARP1 with $0.3\,\mu M$ dumbbell DNA in PARylation Buffer for 15 min with and without $0.5\,mM$ $NAD^+$. ADP-ribose concentration is estimated to be around $355\,\mu M$ by the equation depicted. (C) Urea-PAGE of PAR extracted from mCherry-PARP1 in PARylation reactions with and without $0.5\,mM$ $NAD^+$, labeled with Cy3-dATP. (D) Phase diagram of mCherry-PARP1 condensates formed at the indicated protein and salt concentrations in an autoPARylation reaction with $0.5\,mM$ $NAD^+$. (E) Fluorescence micrograph of mCherry-PARP1 at indicated concentrations with 50 nM triplex DNA and with or without $0.5\,mM$ $NAD^+$. (F) Gel-based autoPARylation assay performed with $1\,\mu M$ mCherry-PARP1 with $0.3\,\mu M$ triplex DNA, with or without $0.5\,mM$ $NAD^+$, and increasing concentrations of ABT-888 (0, 0.01, 0.1, and 1 mM). (G) Fluorescence micrographs of $1\,\mu M$ mCherry-PARP1 with $0.3\,\mu M$ triplex DNA at indicated concentrations of $NAD^+$. (H) Fluorescence micrographs of $1\,\mu M$ mCherry-PARP1 with $0.3\,\mu M$ triplex DNA, with or without $0.5\,mM$ $NAD^+$, and indicated concentrations of ABT-888. (I) Top: Gel-based autoPARylation assay performed with $4\,\mu M$ mCherry-PARP1 with $4\,\mu M$ triplex DNA, with or without $0.5\,mM$ $NAD^+$, and increasing concentrations of PARG. Bottom: Fluorescence micrographs of $4\,\mu M$ mCherry-PARP1 condensates formed in the presence of $4\,\mu M$ triplex DNA, $0.5\,mM$ $NAD^+$, and 125 nM PARG. $N = 1$. (J) Quantifications of the percent of the surface area covered by $4\,\mu M$ mCherry-PARP1 with $4\,\mu M$ triplex DNA with or without $NAD^+$ and HPF1. (K) Fluorescence recovery after photobleaching (FRAP) quantifications of $4\,\mu M$ mCherry-PARP1 with $8\,\mu M$ dumbbell DNA in PARylation Buffer with and without $NAD^+$. All figures represent findings from at least 3 biological replicates in 20 mM Tris pH 7.5, 50 mM NaCl, 7.5 mM $MgCl_2$, and 1 mM DTT unless specified otherwise. Error bars indicate the standard error of the mean. $p$ values are obtained from a student's $t$-test: n.s. non-significant.

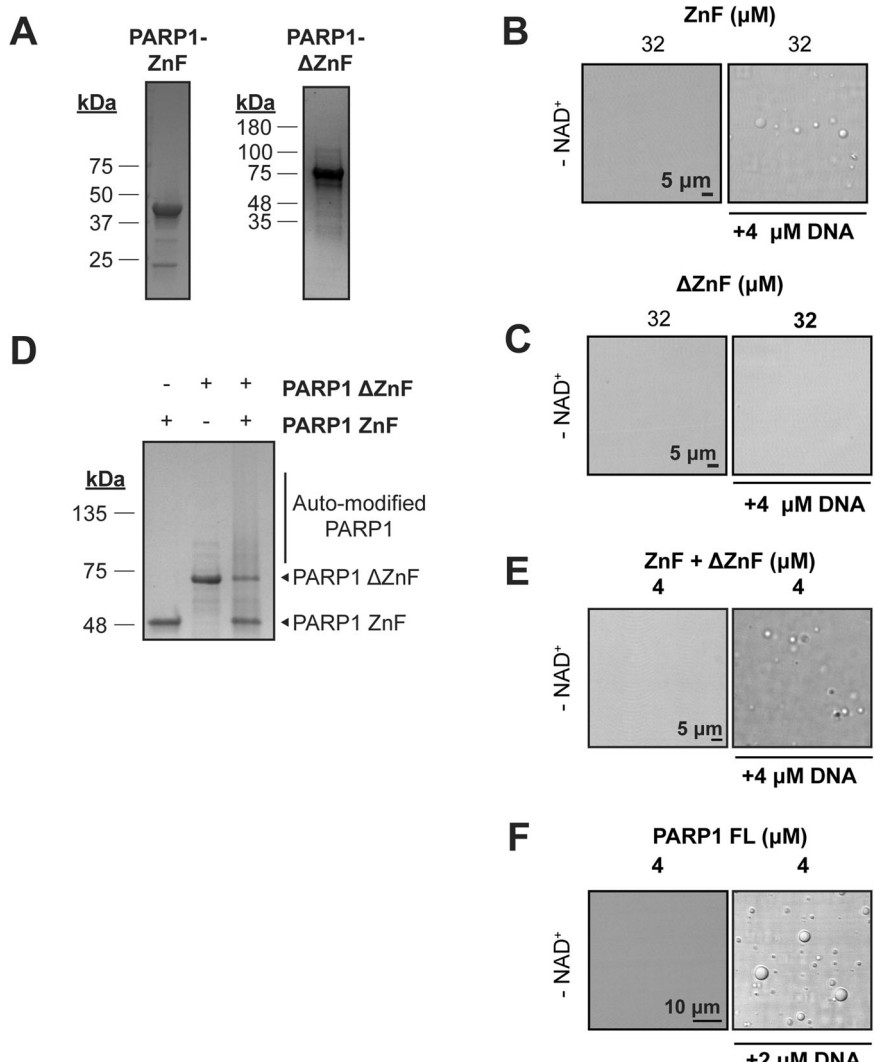

**Figure EV3. The ZnF region is important for PARP1 condensation.**

(A) Coomassie-stained SDS-PAGE gels of purified recombinant PARP1 truncation constructs. (B) DIC micrographs of ZnF PARP1 protein with and without triplex DNA. (C) DIC micrographs of ΔZnF PARP1 protein without triplex DNA. (D) Gel-based autoPARylation assay of 1 μM truncated PARP1 proteins in the presence or absence of 0.5 mM NAD$^+$, analyzed 10 min after NAD$^+$ addition. The absence of a protein band or smearing in assays containing NAD$^+$ indicates modification with PAR. (E) DIC micrographs of ZnF and ΔZnF PARP1 protein fragments mixed together with 4 μM triplex DNA. (F) DIC micrographs of full-length PARP1 protein with and without triplex DNA. All figures represent findings from at least three biological replicates in 20 mM Tris pH 7.5, 50 mM NaCl, 7.5 mM MgCl$_2$, and 1 mM DTT unless specified otherwise.

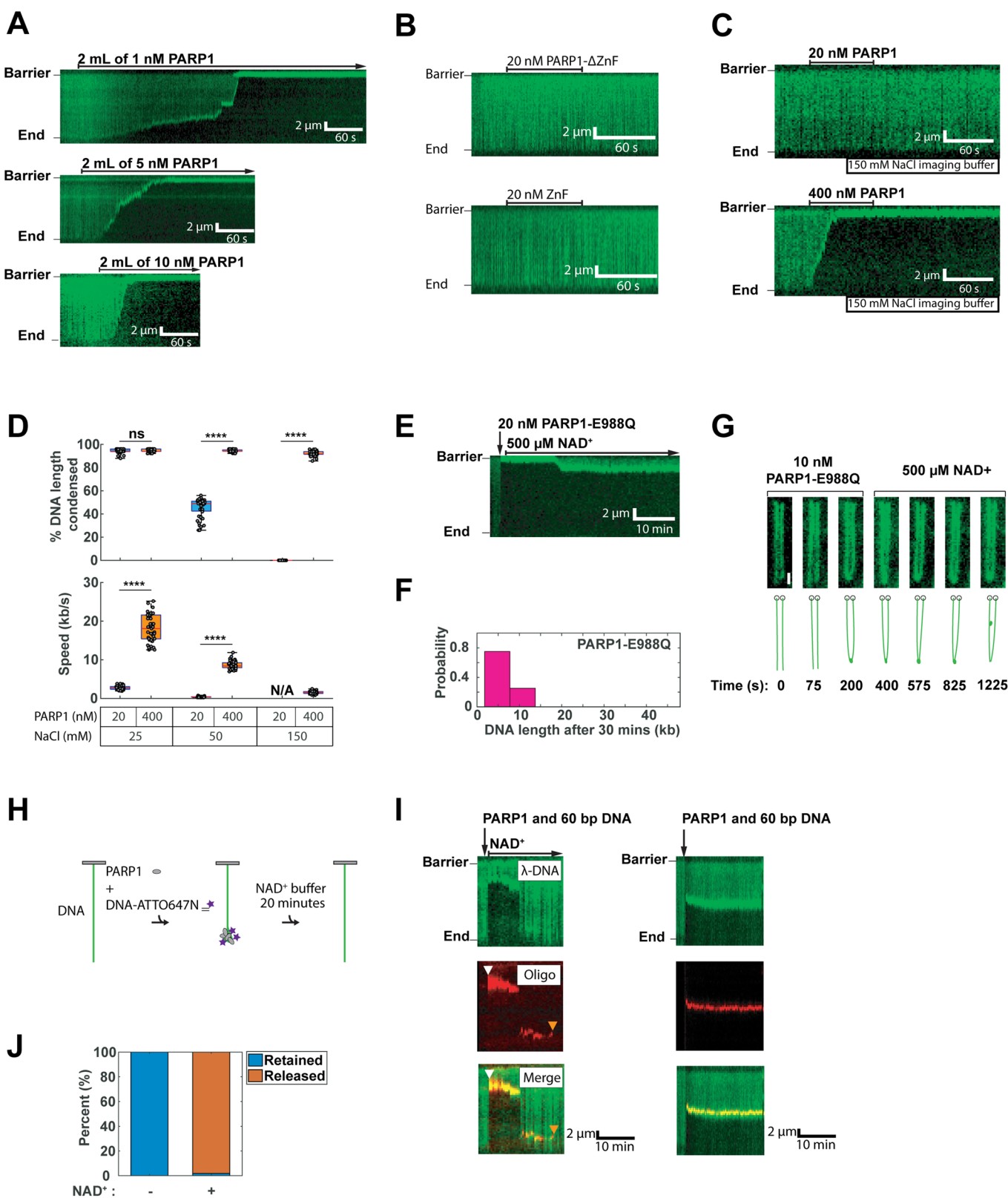

◀ **Figure EV4. PARP1 condensates bridge broken DNA ends.**

(A) Kymographs showing the compaction of DNA (green) induced by the injection of 1, 5, and 10 nM PARP1, respectively. (B) DNA curtains assay with 20 nM of ΔZnF and ZnF mutants of PARP1. Neither mutant can compact DNA at this concentration. (C) Kymographs illustrating DNA compaction by 100 μL of 20 nM PARP1 (top) and 400 nM PARP1 (bottom) with 150 mM NaCl. (D) Quantification of DNA compaction by 100 μL of 20 nM PARP1 or 400 nM PARP1 with 25 mM NaCl, 50 mM NaCl, or 150 mM NaCl. Boxplots denote the first and third quartiles of the data. The center lines in the box indicate the median. The upper whisker is the maximum value and the lower whisker is the minimum value. At least 45 DNA molecules were analyzed for each condition. $p$ values are obtained from a two-tailed $t$-test: ****$p < 0.0001$, ns: not significant. The $p$ value for the condensed DNA length between 20 nM PARP1 and 400 nM PARP1 in 50 mM NaCl buffer condition is 9.1795e-18 and in 150 mM NaCl buffer condition is 4.4296e-19. The $p$ value for the speed between 20 nM PARP1 and 400 nM PARP1 in 25 mM NaCl buffer condition is 5.1175e-17 and in 50 mM NaCl buffer condition is 1.1155e-17. (E) Kymograph showing DNA compaction after injection of 100 μL of 20 nM PARP1-E988Q, followed by DNA extension with buffer containing 500 μM NAD$^+$ for 30 min. (F) Histogram of DNA resolving experiments from E). $N = 51$. (G) Frames from a movie showing the end-to-end bridging of two DNA molecules following the injection of 100 μL of 10 nM PARP1-E988Q. This bridge is not reversed by injecting 500 μM NAD$^+$. Scale bar: 2 μm. (H) Schematic of the 60 bp dsDNA oligo capture and dissociation experiment. (I) Kymographs showing the capture of 60 bp DNA-ATTO647N induced by PARP1. A mixture of 100 μL, containing 5 nM PARP1 and 20 nM DNA-ATTO647N, was injected into the flowcell and incubated with tethered λ-DNA (green) with flow turned off. After a 5-min incubation, imaging acquisition resumed, revealing the capture of DNA-ATTO647N (red) on the long DNA. Subsequently, the imaging buffer was supplemented with (left) or without (right) 500 μM NAD$^+$ buffer. On the left panel, the white arrow marks the time when the dsDNA oligo were captured, while the orange arrow indicates when the dsDNA oligo dissociated following NAD$^+$. (J) Bar graphs depicting the percentage of 60 bp DNA retained and released from λ-DNA following a 20-min chase period with imaging buffer with or without the addition of 500 μM NAD$^+$. Nearly all oligos are released when NAD$^+$ is in the flowcell. At least 57 DNA molecules were analyzed for each condition.

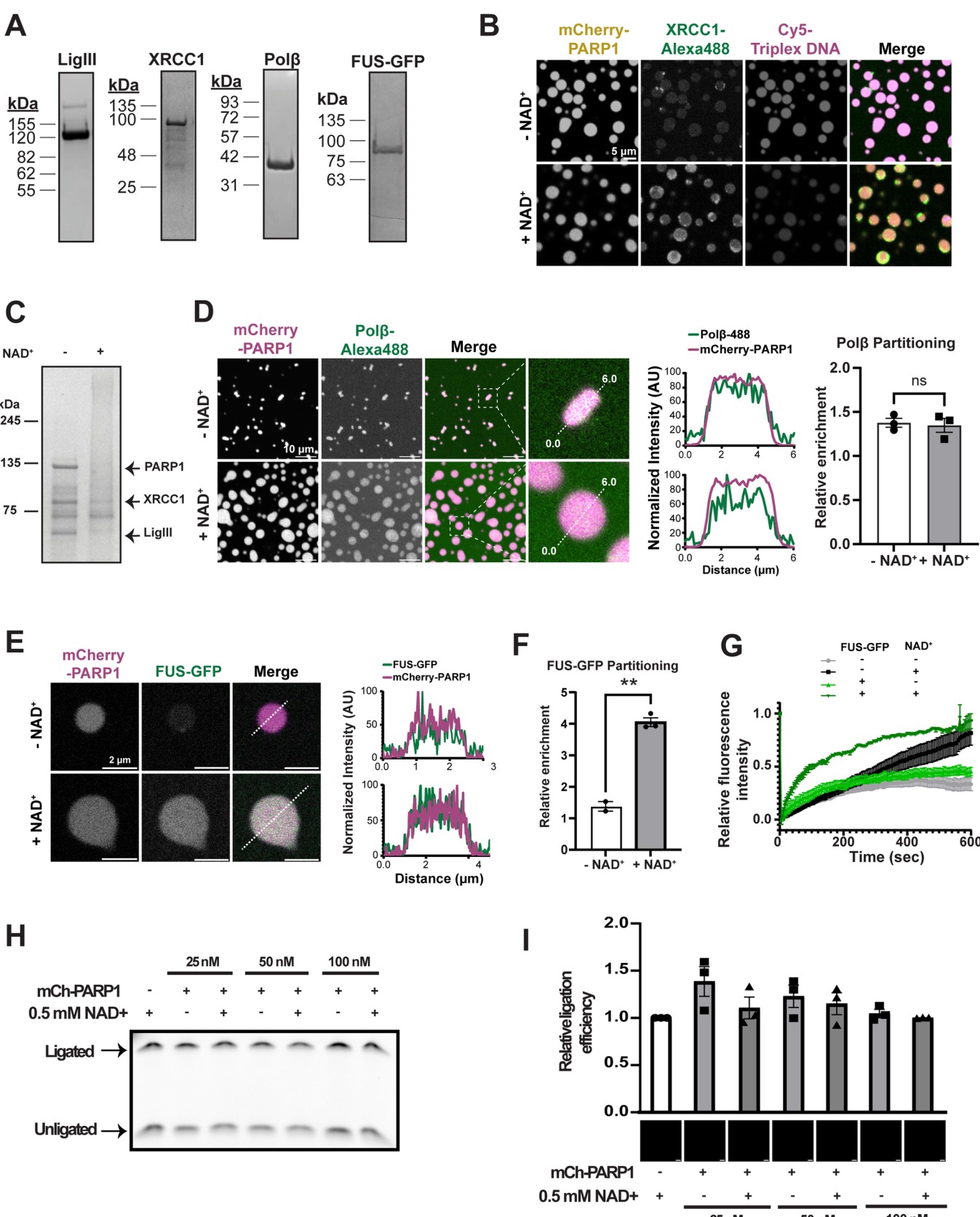

◀ **Figure EV5. SSBR proteins partition together in condensates that enrich DNA.**

(A) Images of Coomassie blue stained SDS-PAGE gels of purified recombinant LigIII, XRCC1, Polβ, and FUS-GFP. (B) Fluorescence micrographs of 4 μM mCherry-PARP1, 4 μM triplex DNA (10% Cy5-labeled), and 1 μM XRCC1 (10% AlexaFluor488-labeled) with and without NAD$^+$. Quantifications of the XRCC1 partitioning are shown in Fig. 5B. (C) Coomassie blue stained SDS-PAGE gel of PARP1, XRCC1, LigIII, triplex DNA, with and without NAD$^+$. Note the disappearance of a strong band at the molecular weight of LigIII, XRCC1, and PARP1 with NAD$^+$, indicating their PARylation. $N = 1$. (D) Right: Fluorescence micrographs of 4 μM mCherry-PARP1, 8 μM dumbbell DNA, and 1 μM Polβ (10% AlexaFluor488-labeled) with and without NAD$^+$. Middle: The fluorescence intensity measured across the diameter of a representative condensate (dashed white line) is plotted on the right. Left: Quantification of Polβ enrichment within condensates, as calculated by the mean fluorescence intensity in the condensed phase relative to that of the dilute phase. (E) Fluorescence micrographs of 4 μM mCherry-PARP1, 1 μM triplex DNA, and 1 μM FUS-GFP with and without NAD$^+$. The fluorescence intensity measured across the white dashed lines is plotted. (F) Quantifications of the FUS-GFP enrichment within PARP1 condensates shown in (E). *p* value = 0.0024. (G) Fluorescence recovery after photobleaching (FRAP) quantifications of 4 μM mCherry-PARP1 with 1 μM triplex DNA with and without 1 μM FUS-GFP in PARylation Buffer with and without NAD$^+$. (H) Gel-based ligation assay was performed with 10 nM LigIII, 10 nM XRCC1, 50 nM Cy5-triplex DNA, and the indicated concentration of mCherry-PARP1 and 0.5 mM NAD$^+$, analyzed 15 min after ATP addition. The presence of a DNA band at a higher molecular weight indicates ligation. (I) Top: Quantifications of ligation efficiency calculated as the fluorescence intensity of Cy5-triplex DNA in the ligated relative to unligated in a gel-based assay shown in (H); Bottom: Representative fluorescence micrographs of mCherry-PARP1 in the ligation reactions after ATP addition. All figures represent findings from at least three biological replicates in 20 mM Tris pH 7.5, 50 mM NaCl, 7.5 mM MgCl$_2$, and 1 mM DTT unless specified otherwise. Error bars indicate the standard error of the mean. *p* values are obtained from a student's *t*-test: **$p < 0.01$, n.s. non-significant.

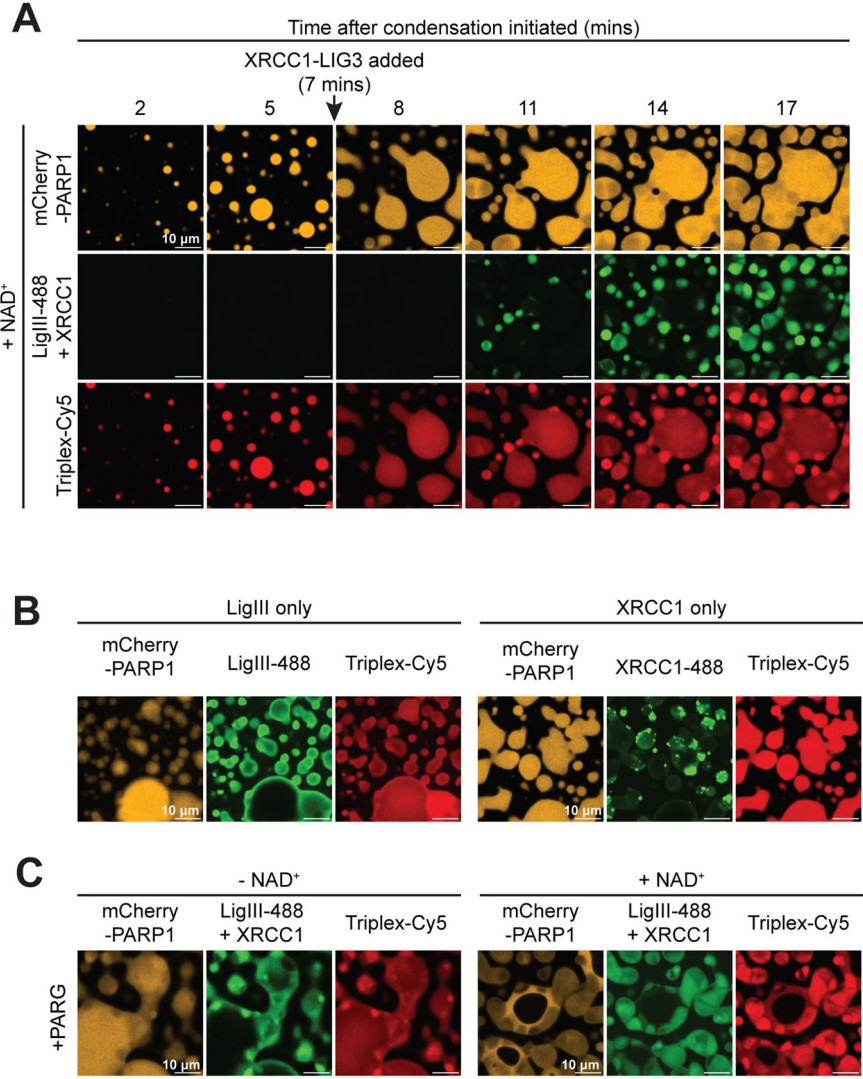

**Figure EV6. SSBR proteins partition together in condensates that enrich DNA.**

(A) Fluorescence micrograph of 4 µM mCherry-PARP1 with 4 µM Triplex DNA and 1 µM XRCC1 and LigIII complex with 0.5 mM NAD$^+$. Imaging started 2 min after PARP1 and DNA were mixed, XRCC1 and LigIII mixture was added 5 min after. (B) Fluorescence micrograph as in A) with LigIII and XRCC1 separately at ~20 min after imaging. (C) Fluorescence micrograph as in (A) with 0.5 µM PARG. PARG was added 5 min after the addition of XRCC1 and LigIII.

