## [Peer Review File · EMBO Reports]

PARP1 condensates differentially partition DNA repair proteins and enhance DNA ligation

Christopher Chin Sang, Gaelen Moore, Maria Tereshchenko, Hongshan Zhang, Michael Nosella, Morgan Dasovich, T. Alderson, Anthony Leung, Ilya Finkelstein, Julie Forman-Kay, and Hyun Lee

Corresponding author(s): Hyun Lee (hyunokate.lee@utoronto.ca)

Review Timeline:

Submission Date:	26th Jul 24
Editorial Decision:	28th Aug 24
Revision Received:	21st Sep 24
Accepted:	1st Oct 24

Editor: Esther Schnapp

**Transaction Report: This manuscript was transferred to
EMBO reports following peer review at Review Commons.**

**Review
COMMONS**

Review #1

1. Evidence, reproducibility and clarity:

Evidence, reproducibility and clarity (Required)

PARP1 plays important roles in the recognition and repair of DNA damage, primarily by catalyzing the formation of PAR at DNA breaks, which assembles various repair factors and other PAR-binding proteins. The anionic PAR scaffold was previously proposed to induce condensation of phase separating proteins including FUS (PMIDs: 26317470, 26286827), and PARP1 was recently reported to form condensates at DNA breaks to promote nucleosome dynamics and DNA end synapsis (PMIDs: 38320550, 38215753). However, several aspects of PARP1-dependent repair condensate formation and how such condensates contribute functionally to repair have remained unclear. Here, Sang et al. show that purified PARP1 forms viscous droplets in vitro in a DNA binding-dependent manner, enhanced by PAR. Interestingly, the downstream DNA repair factors XRCC1, POLB, DNA Ligase III, and FUS co-assemble with PARP1 and damaged DNA, albeit with different enrichment patterns, resulting in multiphase condensates. Functionally, the authors not only confirm a DNA end bridging function by PARP1-mediated condensation, but also report enhanced DNA end ligation. Their in vitro experiments, which are state-of-the-art and are overall well controlled, despite lacking an in vivo counterpart, provide an important step forward in the reconstitution of multi-step DNA repair reactions in repair condensates.

****Major comments:****

1. In addition to the short oligonucleotides that were used in this study to evaluate DNA-dependent PARP1 condensation (Table S2), the use of circular plasmid DNA (nicked or broken to resemble SSBs or DSBs, respectively) should be considered to corroborate key findings.
2. Although not essential for the main conclusions, it would be very interesting to address the role of PARG on PAR-dependent multiphase condensates. Based on the methods section, the authors have purified full-length PARG, so experiments to address the consequences of PARG-dependent PAR degradation on repair condensates and their disassembly seem feasible.
3. If PAR increases the internal dynamics and mobility of PARP1 in condensates, why does it not seem to affect the DNA end bridging function?
4. Catalytically inactive mutants of PARP1 could be employed to separate the PARP1-dependent DNA end bridging function from PAR-dependent modulation of PARP1 dynamics in condensates. Additionally, it would help to show that the DNA end bridging function depends on the ZnF domains and can be modulated by conditions that alter PARP1 condensation (see also point 6).
5. The differential organization of XRCC1, LIG3, POLB, and FUS is intriguing, but the implications of this behavior remain unclear. Can the droplet assays be adapted to inform about the sequence of events from DNA damage recognition by PARP1 and PAR induction to the handing over of the break site to LIG3 for end ligation?
6. The increase in end ligation, which correlates with PARP1- and PAR-dependent condensate formation, is very interesting. However, from the experimental setup it seems unclear if the observed effect is due to condensation or simply PARylation. Additional controls would be needed to substantiate a functional role of condensation for ligation (as implied also in the title of the manuscript). Perhaps it is feasible to modulate condensation (e.g. enhanced by crowding agents,

reduced by salt or by 1,6-hexanediol) without affecting PARylation, and then reassess how this affects ligation.

****Minor comments:****

1. Please double-check if previous studies reaching similar conclusions are referenced appropriately.
2. Please carefully double-check if all references to figure panels are correct.
3. Please carefully double-check if the methods descriptions and discussion match the displayed experimental procedures and results.
4. Supplemental Figure 3 seems to contain only negative results. Consider showing experimental conditions with the same proteins or protein combinations that result in droplet formation as well.

****Referees cross-commenting****

The comments by reviewer #2 seem comprehensible and justified. Similar points are addressed, e.g. major points 2, 3, and 6 in review #1.

2. Significance:

Significance (Required)

Several new and exciting findings on PARP1-dependent and PAR-modulated repair condensate formation are presented, including the multiphase behavior and the functional contribution to DNA compaction, end synopsis, and ligation. The study extends previous work on PARP1/PAR-triggered liquid demixing and complements very new and recently published work by the Alberti and Kay labs on PARP1 condensation at DSBs. The study makes an important step forward towards the reconstitution of DNA repair reactions inside multi-component repair condensates in vitro, which may eventually allow making testable predictions for repair condensate functions in vivo. Strengths of the current study include complementary state-of-the-art in vitro techniques such as biochemical assays, multi-component droplet assays, and single molecule experiments, which were conceived and conducted carefully. Limitations relate primarily to the undemonstrated relevance in cells, which may, however, be beyond the scope of the current study. A broad audience of basic researchers in the areas of genome stability and biomolecular condensates will likely be interested in this study.

3. How much time do you estimate the authors will need to complete the suggested revisions:

Estimated time to Complete Revisions (Required)

(Decision Recommendation)

Between 1 and 3 months

4. Review Commons values the work of reviewers and encourages them to get credit for their work. Select 'Yes' below to register your reviewing activity at Web of Science Reviewer

Recognition Service (formerly Publons); note that the content of your review will not be visible on Web of Science.

No

Review #2

1. Evidence, reproducibility and clarity:

Evidence, reproducibility and clarity (Required)

Sang and colleagues present in vitro evidence for purified PARP1 forming condensates with DNA under certain conditions. They report the PARP1/DNA/NaCl concentration ranges under which these condensates form, and the impact of adding the PARP1 substrate NAD⁺ that allows poly(ADP-ribose) production. The results are presented clearly for the most part. One wonders if these in vitro conditions are really representative of DNA repair foci; however, the study adds new information that will be useful to the field. As noted below, there is some concern about the lack of reversibility of the condensates.

- DNA curtains assay. The imaging buffer is listed as including 25 mM NaCl and 2 mM MgCl₂. Were these experiments performed in conditions of higher ionic strength? The lack of a response to the addition of NAD⁺ is puzzling. It seems that the condensing of the DNA is not reversible. For the data in panel C, would the DNA return to extended form upon further flow of buffer only? The data give the impression that the assay conditions promote a one-way road to an irreversible state, and it is hard to see how this should be interpreted. A different single-molecule study indicated that PARP1 condensation of long DNA is reversible with NAD⁺ addition (PMID 34380612). The different outcomes should be discussed.

- DNA ligase assay. Figure 5E,F. How can it be certain that the ligation is actually taking place in the condensates under these conditions? Can the Cy5 signal be shown? Is there a way to separate the condensates from the dilute phase, and then analyze the ligation state of the DNA? Also, the reactions could be performed under the same conditions (e.g. μ M concentrations of LigIII and XRCC1) as presented in the rest of the figure.

The discussion mentions the impact of HPF1 on PARP1 and that the research team has used HPF1 in the past. Is there a reason for excluding it from the current study? In particular in an effort to address the reversibility/mobility of the condensates?

Supp. Figure 1F. It would be useful to convert the ng/ μ L concentrations to micromolar concentrations, perhaps in the legend for each nucleic acid. This would make the results easier to

relate to the rest of the data that is generally listed in micromolar.

Figure 6A. ZnF3, BRCT, and WGR domains of PARP1 also bind to DNA and should probably be included, as it could help explain why full-length PARP1 is needed for the most robust condensate formation.

Supp. Figure 1G. The legend for this panel indicates absence or presence of NAD⁺, and that the DNA concentration is indicated, but this does not seem consistent with the figure panel.

Results, first sentence. There is a missing parenthesis.

Figure 2G,H. It would be useful to see the plot of the other replicates of the fusion experiments.

page 3. "(data not shown)" Probably worth including.

page 4. "(Supp. Fig. 3B and 3D)" Also Supp. Fig. 3C ?

****Referees cross-commenting****

Reviewer #1 comments are clearly stated and justified. There is good overlap in the feedback.

2. Significance:

Significance (Required)

Strength: the study provides parameters for studying PARP1 condensate formation. Differential impact on repair factors is interesting.

Limitation: the reconstitution is missing elements that could have a very big impact (e.g. nucleosomes).

3. How much time do you estimate the authors will need to complete the suggested revisions:

Estimated time to Complete Revisions (Required)

(Decision Recommendation)

Between 1 and 3 months

4. Review Commons values the work of reviewers and encourages them to get credit for their work. Select 'Yes' below to register your reviewing activity at Web of Science Reviewer

Recognition Service (formerly Publons); note that the content of your review will not be visible on Web of Science.

No

Full Revision

Manuscript number: RC-2024-02397

Corresponding author(s): Hyun O. Lee

1. General Statements [optional]

We thank the reviewers for their valuable feedback on this study. We are thrilled that our work was seen as impactful and agree that the suggested experiments deepen our understanding of how PARP1 condensates are regulated and function. In the revised manuscript, we include results from all of the experiments suggested by the reviewers and feel that the manuscript is further strengthened as a result. Below, we summarize our findings point-by-point.

Reviewer #1 (Evidence, reproducibility and clarity (Required)):

PARP1 plays important roles in the recognition and repair of DNA damage, primarily by catalyzing the formation of PAR at DNA breaks, which assembles various repair factors and other PAR-binding proteins. The anionic PAR scaffold was previously proposed to induce condensation of phase separating proteins including FUS (PMIDs: 26317470, 26286827), and PARP1 was recently reported to form condensates at DNA breaks to promote nucleosome dynamics and DNA end synapsis (PMIDs: 38320550, 38215753). However, several aspects of PARP1-dependent repair condensate formation and how such condensates contribute functionally to repair have remained unclear. Here, Sang et al. show that purified PARP1 forms viscous droplets in vitro in a DNA binding-dependent manner, enhanced by PAR. Interestingly, the downstream DNA repair factors XRCC1, POLB, DNA Ligase III, and FUS co-assemble with PARP1 and damaged DNA, albeit with different enrichment patterns, resulting in multiphase condensates. Functionally, the authors not only confirm a DNA end bridging function by PARP1-mediated condensation, but also report enhanced DNA end ligation. Their in vitro experiments, which are state-of-the-art and are overall well controlled, despite lacking an in vivo counterpart, provide an important step forward in the reconstitution of multi-step DNA repair reactions in repair condensates.

We thank the reviewer for this positive review.

Major comments:

1) In addition to the short oligonucleotides that were used in this study to evaluate DNA-dependent PARP1 condensation (Table S2), the use of circular plasmid DNA (nicked or broken to resemble SSBs or DSBs, respectively) should be considered to corroborate key findings.

Thank you for this suggestion. Our new Supp. Fig. 1 now includes results with circular plasmids with and without a single nick (**Fig. 1I** and **Supp. Fig. 1E**). We find that both undamaged and nicked plasmid DNA induce PARP1 phase separation to a largely similar extent. Based on our results with 20-200bp dsDNA fragments, the length of the DNA is an important factor in PARP1 condensation. From this, we conclude that a weak binding of PARP1 to undamaged, very long DNA (2.7kb) is sufficient to drive condensate formation. We hypothesize that the effect of a single nick is masked by the length of the plasmid DNA, leading to a largely similar extent of PARP1 condensation.

2) Although not essential for the main conclusions, it would be very interesting to address the role of PARG on PAR-dependent multiphase condensates. Based on the methods section, the authors have purified full-length PARG, so experiments to address the consequences of PARG-dependent PAR degradation on repair condensates and their disassembly seem feasible.

We agree that this is an interesting question and have now included several new results from experiments with PARG. In our first submission, we focused on the predicted concentration of PARP1 in the nucleus, which is 1 μ M or higher (Hein et al. 2015). At these concentrations, neither condensation nor multiphase formation with other DNA repair proteins (namely XRCC1 and LigIII) were dependent on PARylation. Consistent with this, we find that the addition of PARG did not dissolve PARP1 condensates, although it did reduce their size back to non-PARylated condition (**Fig. 2E and 2F**, **Supp. Fig. 2I**), and did not affect multiphase behavior at these PARP1 concentrations (**Supp. Fig. 6C**).

3) If PAR increases the internal dynamics and mobility of PARP1 in condensates, why does it not seem to affect the DNA end bridging function?

Thank you for this idea. In the previous submission, we had pulsed in 0.5 mM NAD⁺ buffer into the system for a short time period. We now include conditions with varied concentrations of NAD⁺ over longer periods of time, and thus more extensive PARylation. Longer exposure of NAD⁺ reversed DNA compaction by PARP1, which was accelerated by increasing NAD⁺ concentration (**Fig. 4E and F**). Similarly, DNA end-bridging by PARP1 was reversed by prolonged addition of NAD⁺ (**Fig. 4G**).

4) Catalytically inactive mutants of PARP1 could be employed to separate the PARP1-dependent DNA end bridging function from PAR-dependent modulation of PARP1 dynamics in condensates. Additionally, it would help to show that the DNA end bridging function depends on the ZnF domains and can be modulated by conditions that alter PARP1 condensation (see also point 6).

Thank you for this suggestion. Our new Fig. 4 and Supp. Fig. 4 include several new experiments to characterize PARP1-dependent DNA compaction and end-bridging and the impact of PARylation on these functions. For example, we include results using PARP1 E988Q, a mutant deficient in PARylation but proficient in MARYlation, in our single-molecule DNA curtain experiment. This mutant was unable to reverse DNA compaction and bridge DNA ends upon addition of NAD⁺ (**Supp. Fig. 4E - G**), unlike wildtype PARP1 (**Fig. 4E - G**), indicating that the end bridging function of PARP1 condensates does not require autoPARylation but PARylation can reverse this. The mutant missing the ZnF domains (Δ ZnF) was unable to condense DNA, indicating that ZnF region is necessary for this function (**Supp. Fig. 4B**), and thus likely will not be able to bridge DNA. We also observe that increasing the salt concentration to 150 mM, which dissolves pre-existing PARP1 condensates (**Supp. Fig. 1F**), also prevented DNA compaction at low PARP1 concentrations (**Supp. Fig. 4C**), although higher PARP1 concentration could still induce DNA compaction at 150mM salt. Together, DNA compaction and end-bridging functions closely correlate with conditions that lead to PARP1 condensation and are disrupted by conditions that dissolve them or increase their dynamics such as high salt and PARylation.

5) The differential organization of XRCC1, LIG3, POLB, and FUS is intriguing, but the implications of this behavior remain unclear. Can the droplet assays be adapted to inform about the sequence of events from DNA damage recognition by PARP1 and PAR induction to the handing over of the break site to LIG3 for end ligation?

Thank you for this suggestion. To understand this, we monitored the organization of PARP1, LigIII/XRCC1, and DNA over time with and without NAD⁺. In this experiment, we pre-formed mCherry-PARP1 condensates by adding DNA, then added LigIII/XRCC1 several minutes after. We found that the way PARP1 and LigIII/XRCC1 interact depends upon the presence of NAD⁺. Upon the addition of LigIII/XRCC1, we observed a decrease in the DNA signal in mCherry-PARP1 condensates and a subsequent increase in DNA partitioning to the LigIII/XRCC1 phase, reminiscent of a hand-off of the DNA break site from PARP1 to LigIII for ligation. However, the preferential partitioning of DNA to LigIII/XRCC1 was unaffected by the presence of NAD⁺, indicating that the handing over of the break site is driven by DNA's preferential interactions with LigIII/XRCC1 rather than the PARP1 PARylation state. These data are now included in the new **Figure 5** and **Supp. Figure 6**.

6) The increase in end ligation, which correlates with PARP1- and PAR-dependent condensate formation, is very interesting. However, from the experimental setup it seems unclear if the observed effect is due to condensation or simply PARylation. Additional controls would be needed to substantiate a functional role of condensation for ligation (as implied also in the title of the manuscript). Perhaps it is feasible to modulate condensation (e.g. enhanced by crowding agents, reduced by salt or by 1,6-hexanediol) without affecting PARylation, and then reassess how this affects ligation.

Thank you for this suggestion and we agree that teasing apart the contribution of PARP1- and PAR-dependent condensate formation would be a valuable addition.

To this end, we tried several approaches but did not find a condition to modulate PARP1 condensation without affecting PARylation activity. We found that the crowding agents we tested did not enhance PARP1 condensation. The presence of up to 10% 1,6-hexanediol in the reaction did not dissolve PARP1 condensates, while reducing the size of PARP1 condensates (**Supp. Fig 1G**), consistent with the model that PARP1 condensation is driven by electrostatic interactions. Dissolving pre-formed PARP1 condensates with higher salt concentrations (50 to 250 mM) also significantly impaired the PARylation activity of PARP1. Increased salt has also been shown to impair DNA binding (Langelier et al. 2011, Clark et al. 2012, Langelier et al. 2018). Lastly, we aimed to separate PARP1 condensation from autoPARylation using a PARylation-deficient but MARYlation-proficient mutant, PARP1 E988Q. This mutant consistently led to significant retention of DNA in the wells of our denaturing gels that could not be resolved with increased denaturing agents or with protease treatment. This may suggest that PARP1 E988Q binds DNA more strongly than the wildtype as previously reported (Shao et al. 2020). Accordingly, DNA ligation efficiency could not be quantified accurately.

As the effect of PARP1 condensates with and without PARylation on ligation could be compared directly, we measured the effect of PARylation at PARP1 concentrations that do not allow for condensate formation even after PARylation (25, 50, 100 nM). In all conditions tested, we did not observe any significant differences in the ligation efficiency with PARylation (**Supp. Fig. 5H and 5I**). Thus, we conclude that the increased ligation efficiency observed in the presence of PARP1 condensates induced by PARylation is due to the presence of PARP1 condensates.

Minor comments:

1) Please double-check if previous studies reaching similar conclusions are referenced appropriately.

We have now updated and expanded the discussion of relevant studies.

2) Please carefully double-check if all references to figure panels are correct.

Full Revision

Thank you for noticing this, we have checked that all figure callouts are correct.

3) Please carefully double-check if the methods descriptions and discussion match the displayed experimental procedures and results.

Thank you. This has been done.

4) Supplemental Figure 3 seems to contain only negative results. Consider showing experimental conditions with the same proteins or protein combinations that result in droplet formation as well.

Thank you for this suggestion. In the original submission, we had moved the control experiments that did not show condensate formation in the supplement to simplify the main figure that contained the conditions that result in droplet formation. To facilitate comparison, we now refer to associated results showing condensation formation in **Supp. Fig. 3** and show phase diagrams wherever appropriate.

****Referees cross-commenting****

The comments by reviewer #2 seem comprehensible and justified. Similar points are addressed, e.g. major points 2, 3, and 6 in review #1.

Reviewer #1 (Significance (Required)):

Several new and exciting findings on PARP1-dependent and PAR-modulated repair condensate formation are presented, including the multiphase behavior and the functional contribution to DNA compaction, end synapsis, and ligation. The study extends previous work on PARP1/PAR-triggered liquid demixing and complements very new and recently published work by the Alberti and Kay labs on PARP1 condensation at DSBs. The study makes an important step forward towards the reconstitution of DNA repair reactions inside multi-component repair condensates in vitro, which may eventually allow making testable predictions for repair condensate functions in vivo. Strengths of the current study include complementary state-of-the-art in vitro techniques such as biochemical assays, multi-component droplet assays, and single molecule experiments, which were conceived and conducted carefully. Limitations relate primarily to the undemonstrated relevance in cells, which may, however, be beyond the scope of the current study. A broad audience of basic researchers in the areas of genome stability and biomolecular condensates will likely be interested in this study.

Reviewer #2 (Evidence, reproducibility and clarity (Required)):

Sang and colleagues present in vitro evidence for purified PARP1 forming condensates with DNA under certain conditions. They report the PARP1/DNA/NaCl concentration ranges under which these condensates form, and the impact of adding the PARP1 substrate NAD⁺ that allows poly(ADP-ribose) production. The results are presented clearly for the most part. One wonders if these in vitro conditions are really representative of DNA repair foci; however, the study adds new information that will be useful to the field. As noted below, there is some concern about the lack of reversibility of the condensates. Thank you for this positive review.

##

DNA curtains assay. The imaging buffer is listed as including 25 mM NaCl and 2 mM MgCl₂. Were these experiments performed in conditions of higher ionic strength? The lack of a response to the addition of NAD⁺ is puzzling. It seems that the condensing of the DNA is not reversible. For the data in panel C, would the DNA return to extended form upon further flow of buffer only? The data give the impression that the assay conditions promote a one-way road to an irreversible state, and it is hard to see how this should be interpreted. A different single-molecule study indicated that PARP1 condensation of long DNA is reversible with NAD⁺ addition (PMID 34380612). The different outcomes should be discussed.

Thank you for these suggestions. In our first submission, we focused on the DNA compaction under lower ionic strength where PARP1 more readily forms condensates. We now include results at higher salt concentrations (150 mM NaCl), which prevents DNA compaction at 20 nM (Supp. Fig. 4C). However, this was PARP1 concentration dependent, with 150mM NaCl unable to reverse DNA compaction at 400 nM PARP1. Thus, DNA compaction is sensitive to ionic strength of the buffer and PARP1 concentration.

Previously, we focused on a short pulse of low concentration of NAD⁺. We now include results from longer pulses and higher concentrations of NAD⁺ that show that DNA compaction can be reversed in NAD⁺ dependent manner (Fig. 4E and 4F). Our data agrees with findings by Bell et al. 2021 and our discussion on this has also been expanded.

##

DNA ligase assay. Figure 5E,F. How can it be certain that the ligation is actually taking place in the condensates under these conditions? Can the Cy5 signal be shown? Is there a way to separate the condensates from the dilute phase, and then analyze the ligation

state of the DNA? Also, the reactions could be performed under the same conditions (e.g. μM concentrations of LigIII and XRCC1) as presented in the rest of the figure.

Thank you for raising this point and for the suggested experiments.

Cy5 signal for the ligation assay condition has now been included in **Fig. 5E**, demonstrating enrichment of the DNA substrate in PARP1 condensates as in **Fig. 4A**. Unfortunately, because Cy5 is conjugated to one of the two short DNA sequences annealed to longer DNA to make up the nicked DNA triplex with two blunt ends in our assays, Cy5-signal could not be used to distinguish between ligated and unligated DNA.

The conditions presented in the rest of the figure, optimized for visualizing the organization within the condensates, is unfortunately beyond the dynamic range of the ligation assay we used – ligation detection plateaus at $1 \mu\text{M}$ LigIII leading to 100% ligation in all conditions. Therefore, we used $0.01 \mu\text{M}$ LigIII, which allowed us to differentiate changes to ligation efficiency between the conditions.

In both the microscopy and ligation conditions, PARP1 condensates represent a fraction of the total volume, and we were not able to conduct multiple experiments with condensed and dilute phases separated.

With these limitations in mind, we focused on understanding the role of PARP1 and PARylation as follows. In our assays, the concentration of DNA ligase III, XRCC1, and DNA remains constant between the conditions. Thus, we conclude that the differences in ligation we detect is likely due to the presence of PARP1 or the organization by PARP1 condensates. To test the impact of PARP1 with or without PARylation in these assays, we conducted several experiments that detect ligation when 1) PARP1 condensates are dissolved; 2) PARP1 forms comparable amount of condensates with and without PARylation; 3) PARP1 is not condensed with or without NAD^+ . We did not find a condition where dissolving PARP1 does not influence PARylation (1). PARylation consistently increases the condensed volume fraction and thus were not equivalent to non-PARylated conditions (2) - please also see our response to reviewer 1 point 6. By lowering the PARP1 concentration, we addressed 3) and observed no difference in ligation efficiency upon PARylation (**Supp. Fig. 5H and 5I**). From this, we conclude that rather than the presence of PARP1 or PARylation, the organization of PARP1 into condensates promotes efficient ligation.

##

The discussion mentions the impact of HPF1 on PARP1 and that the research team has used HPF1 in the past. Is there a reason for excluding it from the current study? In particular in an effort to address the reversibility/mobility of the condensates?

Thank you for suggesting this. We have now added results from experiments with HPF1 to address its effect on PARP1 condensation reversibility to **Supp. Fig. 2**. Briefly, addition of HPF1 did not reverse PARP1 condensates (**Supp. Fig. 2J**). We have included these findings in the results and updated the discussion accordingly.

##

Supp. Figure 1F. It would be useful to convert the ng/uL concentrations to micromolar concentrations, perhaps in the legend for each nucleic acid. This would make the results easier to relate to the rest of the data that is generally listed in micromolar.

Thank you for this suggestion. We have updated DNA concentrations to indicate both ng/ μ L and μ M for easy comparison.

##

Figure 6A. ZnF3, BRCT, and WGR domains of PARP1 also bind to DNA and should probably be included, as it could help explain why full-length PARP1 is needed for the most robust condensate formation.

Thank you for the suggestion. We have added this to our schematic.

##

Supp. Figure 1G. The legend for this panel indicates absence or presence of NAD⁺, and that the DNA concentration is indicated, but this does not seem consistent with the figure panel.

Thank you for noticing this. During the revision, some of the figure panels have moved. However, the figure legend for previous Fig 1G has now been corrected. We found that NAD⁺ does not alter the phase diagram of PARP1 at the protein and DNA concentrations in this phase diagram in our experiments and thus the phase diagram for both the absence and presence of NAD⁺ are the same. This has also been noted in the figure legend for clarity.

##

Results, first sentence. There is a missing parenthesis.

Thank you for pointing this out. This has now been corrected.

##

Figure 2G,H. It would be useful to see the plot of the other replicates of the fusion experiments.

Thank you for this suggestion. We have now included the aspect ratio from 5 fusion events for each condition and plotted the average with the error bars depicting the standard error of the mean (**Fig. 2J**). This updated graph is consistent with our previous conclusion that PARylated PARP1 condensates relax into a spherical shape faster than their unPARylated counterparts.

##

page 3. "(data not shown)" Probably worth including.

We agree. The fluorescence recovery after photobleaching (FRAP) data of mCherry-PARP1 with the dumbbell DNA are now included in **Supp. Fig. 2K**. This result shows similar results as obtained with triplex DNA; PARylation increases the internal dynamics of PARP1 condensates.

##

page 4. "(Supp. Fig. 3B and 3D)" Also Supp. Fig. 3C?

Thank you for noticing this. The figure reference has now been corrected accordingly.

****Referees cross-commenting****

Reviewer #1 comments are clearly stated and justified. There is good overlap in the feedback.

Reviewer #2 (Significance (Required)):

Strength: the study provides parameters for studying PARP1 condensate formation. Differential impact on repair factors is interesting.

Limitation: the reconstitution is missing elements that could have a very big impact (e.g. nucleosomes).

Thank you for raising this point. We agree that it is important to understand how PARP1 condensate formation and function are influenced by nucleosome-bound DNA. In our recent publication (Nosella et al. 2023), we described that PARP1 and nucleosomes form condensates together. We feel the current study provides a crucial foundation for understanding this behavior and how nucleosome-free DNA and PAR regulate PARP1 condensate formation. Together, these results contribute important insights for future studies on the organization and activity of DNA repair foci.

Dear Dr. Lee,

Thank you for the submission of your revised manuscript. We have now received the enclosed reports from the referees that were asked to assess it. Referee 1 still has a few more minor suggestions that I would like you to incorporate before we can proceed with the official acceptance of your manuscript.

A few editorial requests will also need to be addressed:

- Please upload a ms word file without figures with your final submission.
- Please reduce the number of keywords to 5.
- Please add a Data Availability Section (DAS) to the end of the Methods that lists accession numbers for data generated in this study and deposited in public databases. If no such data were deposited, please mention this fact in the DAS.
- Please correct the conflict of interest subheading to "Disclosure and Competing Interests Statement"
- The author credits need to be removed from the ms file. All credits need to be entered during online ms submission.
- The reference format needs to be corrected to the EMBO reports (Harvard) style. It needs to be alphabetical, not numerical; et al needs to be used after 10 author names.
- Please send us a completed author checklist, which you can download from our author guidelines <<https://www.embopress.org/page/journal/14693178/authorguide>>. The completed author checklist will also be part of the transparent peer-review process file.
- Several funding information is missing in our online ms submission system. Please enter all funding info also there with your final submission.
- Please upload all main and all EV figures as individual, high resolution figure files with your final ms.
- Please add currently missing callouts for Fig 4D and Fig. 6A.
- The suppl. figures and tables in the ms need to be removed from the ms and placed in a separate PDF file titled Appendix with their legends; the file must have a table of content with page numbers on the title page and the correct nomenclature and ms callouts should be Appendix Table S1-S2, Appendix Figure S1-S6. Alternatively, we can accommodate these figures and tables as EV figures and tables (which should be upldd as separate Figure files and their legends should be in the ms, after the main figure legends) that are imbedded in the ms text online. You can find more information about our file types in our guide to authors online.
- The movies need to be renamed to Movie EV1-EV3 (as do the ms callouts); each movie should have a corresponding legend provided as a readme.txt file and each movie should be zipped up together with its legend and upldd as one folder per movie.
- Since July, all our papers need to include a Reagents and Tools Table. You can download a template (.docx) for the Reagents and Tools Table from our author guidelines: <<https://www.embopress.org/page/journal/14693178/authorguide#manuscriptpreparation>>.
- The manuscript sections should be in the following order: Title page - Abstract & Keywords - Introduction - Results - Discussion - Methods - Data Availability - Acknowledgments - Disclosure Statement & Competing Interests - References - Figure Legends - (Main Tables with legends) - Expanded View Figure Legends.
- Our image check of all ms figures found a Cell re-use between Figure 3E Panel 4 and Appendix Figure 3 E and F , which is not detailed in the figure legend. Please explain this image re-use. If it is intentional, please mention the re-use in the figure legends.
- Please note that the supplementary figure 1 does not contain any quantification graph, kindly rectify the statistics related information in the figure legend appropriately.
- Please define the annotated p values ** as well as provide the exact p-values for the same in the legend of figure 2b; as appropriate.
- Please note that the exact p values are not provided in the legends of figures 4a, d; 5a-c, e; supplementary figures 4d; 5f.

- Please indicate the statistical test used for data analysis in the legend of figure 2b.
- Please note that in supplementary figures 5f-g, i; there is a mismatch between the annotated p values in the figure legend and the annotated p values in the figure file that should be corrected.
- Please note that for the supplementary figures 2j-k, p-values and statistical tests are indicated in the legends. However, comparison for the same, "***/*" has not been represented in the figures. Please rectify this in the figures or legends as applicable.
- Please note that the box plots need to be defined in terms of minima, maxima, centre, bounds of box and whiskers, and percentile in the legends of figure 4d, supplementary figure 4d.
- Although 'n' is provided, please describe the nature of entity for 'n' in the legend of figure 4a.
- Please note that the error bars are not defined in the legend of figure 4a
- Please note that the scale bar needs to be defined for figure 5f, supplementary figure 6a-c.
- Please note that scale bar and its definition are missing for supplementary figure 2i.
- Please note that the white/orange arrowheads are not defined in the legend of supplementary figure 4i. This needs to be rectified.

Regarding the abstract, it might be good to add that all experiments are performed in vitro (if I remember correctly).

EMBO press papers are accompanied online by A) a short (1-2 sentences) summary of the findings and their significance, B) 2-3 bullet points highlighting key results and C) a synopsis image that is exactly 550 pixels wide and 200-600 pixels high (the height is variable). The synopsis image should provide a sketch of the major findings, like a graphical abstract. Please note that text needs to be readable at the final size. Please send us this information along with the final manuscript.

Referee #1:

The revised manuscript now includes several additional informative results and control conditions, overall supporting the main conclusions. I endorse publication in EMBO Reports pending a few small textual changes:

- To avoid confusion between in vitro findings on condensate formation and cellular DNA repair foci, I recommend two small modifications to the abstract: Change to: "... it remains unclear how exactly PARP1 and PARylation ..." and change to: "These findings provide insights into how PARP1 condensation and PARylation regulate the assembly and biochemical activities of DNA repair factors, which may inform on how PARPs function in DNA repair foci and other PAR-driven condensates."
- Page 6, last line, it might be clearer to write: ", ... rather than the presence of PARP1 or PARylation alone, ...".
- Discussion, first sentence, it would seem more appropriate to write: "In this study, we examine the role of PARP1 and PARylation in the formation, organization, and function of nascent, multi-component DNA repair condensates in vitro". In the following sentence, removing the subclause seems appropriate considering the previous literature ("...", thus expanding ...").
- Discussion, page 8 first paragraph, could be an opportunity to briefly discuss what the implications of different DNA substrates might be for nuclear PARP1 functions and PAR-dependent condensate formation, i.e. which type(s) of DNA damage and which cellular conditions would be predicted to strongly induce PAR-dependent condensation, and which ones perhaps not?
- Figure legend to Figure 6B, please consider including references for the reported interactions between the proteins and their different domains.

Referee #2:

In the resubmitted version of this manuscript, the authors have carefully addressed my comments on the initial submission with additional experimentation and changes to the text. I view the re-submission as ready for publication.

We would like to thank the reviewers for their thoughtful and positive feedback on our study. We have incorporated all suggestions in the revised version and have highlighted sections that were updated in the manuscript (please see 'manuscript with highlights').

Referee #1:

The revised manuscript now includes several additional informative results and control conditions, overall supporting the main conclusions. I endorse publication in EMBO Reports pending a few small textual changes:

We thank the reviewer for this positive review.

- To avoid confusion between in vitro findings on condensate formation and cellular DNA repair foci, I recommend two small modifications to the abstract: Change to: "... it remains unclear how exactly PARP1 and PARylation ..." and change to: "These findings provide insights into how PARP1 condensation and PARylation regulate the assembly and biochemical activities of DNA repair factors, which may inform on how PARPs function in DNA repair foci and other PAR-driven condensates."

Thank you for your suggestion. We have incorporated the suggested changes and additional wording to the abstract to clarify that our conclusions are from in vitro work.

- Page 6, last line, it might be clearer to write: ", ... rather than the presence of PARP1 or PARylation alone, ...".

We agree and have incorporated your suggestion.

- Discussion, first sentence, it would seem more appropriate to write: "In this study, we examine the role of PARP1 and PARylation in the formation, organization, and function of nascent, multi-component DNA repair condensates in vitro". In the following sentence, removing the subclause seems appropriate considering the previous literature ("...", thus expanding ...").

Thank you, we have changed the sentences in the discussion accordingly.

- Discussion, page 8 first paragraph, could be an opportunity to briefly discuss what the implications of different DNA substrates might be for nuclear PARP1 functions and PAR-dependent condensate formation, i.e. which type(s) of DNA damage and which cellular conditions would be predicted to strongly induce PAR-dependent condensation, and which ones perhaps not?

We appreciate your suggestion. We've added a few lines to tackle this topic in the discussion.

- Figure legend to Figure 6B, please consider including references for the reported interactions between the proteins and their different domains.

Thank you for this suggestion. We have added a few references for each interaction depicted in Figure 6B in the legend.

Referee #2:

In the resubmitted version of this manuscript, the authors have carefully addressed my comments on the initial submission with additional experimentation and changes to the text. I view the re-submission as ready for publication.

Thank you for this positive review.

Hyun Lee
University of Toronto
Canada

Dear Dr. Lee,

I am very pleased to accept your manuscript for publication in the next available issue of EMBO reports. Thank you for your contribution to our journal.
